# Symmetry-breaking in patch formation on triangular gold nanoparticles by asymmetric polymer grafting

Ahyoung Kim [1,11], Thi Vo[2,11], Hyosung An[1,9], Progna Banerjee [3,4], Lehan Yao [1], Shan Zhou [1,10], Chansong Kim [1], Delia J. Milliron [3], Sharon C. Glotzer [2,5] ✉ & Qian Chen [1,6,7,8] ✉

Synthesizing patchy particles with predictive control over patch size, shape, placement and number has been highly sought-after for nanoparticle assembly research, but is fraught with challenges. Here we show that polymers can be designed to selectively adsorb onto nanoparticle surfaces already partially coated by other chains to drive the formation of patchy nanoparticles with broken symmetry. In our model system of triangular gold nanoparticles and polystyrene-*b*-polyacrylic acid patch, single- and double-patch nanoparticles are produced at high yield. These asymmetric single-patch nanoparticles are shown to assemble into self-limited patch–patch connected bowties exhibiting intriguing plasmonic properties. To unveil the mechanism of symmetry-breaking patch formation, we develop a theory that accurately predicts our experimental observations at all scales—from patch patterning on nano-particles, to the size/shape of the patches, to the particle assemblies driven by patch–patch interactions. Both the experimental strategy and theoretical prediction extend to nanoparticles of other shapes such as octahedra and bipyramids. Our work provides an approach to leverage polymer interactions with nanoscale curved surfaces for asymmetric grafting in nanomaterials engineering.

Patchy particles[1–4] have attracted increasing synthetic attention for applications in directed assembly[5–9], plasmonics[10], catalysis[11], and targeted delivery[12] thanks to their directional interaction[1,2], hybrid composition[13], and force imbalance under external fields[14]. Previous synthetic efforts to induce polymeric patch formation on anisotropic nanoparticles (NPs) rely on NP shape variations to drive post-synthesis

segregation of uniformly adsorbed polymers[15–17], spontaneous phase separation of ligand mixtures[18], partial ligand exchange[19], or curvature-induced selective adsorption of polymers[20–22]. However, such patch formations are governed by NP geometry, making it difficult to target specific surface facets/locations. Here we report an approach to engineer patches on anisotropic NPs through the grafting of polymers

[1]Department of Materials Science and Engineering, University of Illinois, Urbana, IL 61801, USA. [2]Department of Chemical Engineering, University of Michigan, Ann Arbor, MI 48109, USA. [3]Center for Nanoscale Materials, Nanoscience and Technology Division, Argonne National Laboratory, Lemont, IL 60439, USA. [4]McKetta Department of Chemical Engineering, The University of Texas at Austin, Austin, TX 78712, USA. [5]Biointerfaces Institute, University of Michigan, Ann Arbor, MI 48109, USA. [6]Department of Chemistry, University of Illinois, Urbana, IL 61801, USA. [7]Beckman Institute for Advanced Science and Technology, University of Illinois, Urbana, IL 61801, USA. [8]Materials Research Laboratory, University of Illinois, Urbana, IL 61801, USA. [9]Present address: Department of Petrochemical Materials Engineering, Chonnam National University, Yeosu 59631, Korea. [10]Present address: Department of Nanoscience and Biomedical Engineering, South Dakota School of Mines and Technology, Rapid City, SD 57701, USA. [11]These authors contributed equally: Ahyoung Kim, Thi Vo. ✉e-mail: sglotzer@umich.edu; qchen20@illinois.edu

onto curved NP surfaces. We show that polymer–polymer attraction actively recruits free chains in solution to adsorb onto the surface regions of NPs already partially occupied by other polymers, overcoming competing energetic costs such as polymer-solvent attraction and steric hinderance[23]. This process ultimately leads to asymmetric polymer adsorption that can be leveraged to synthesize patchy NPs with broken symmetry. In contrast to other synthesis methods, our method utilizes asymmetric molecular adsorption via polymer–polymer interaction to induce symmetry breaking during the grafting process, overwriting NP intrinsic shape symmetry. We present a scaling theory that can a priori predict both morphological and structural details of polymeric patches on NPs. Our theory can aid in future experimental designs of asymmetric patchy particles that leverages the rich chemical/structural diversity, size tunability, and conformational entropy[24] intrinsic to polymers[25].

We use a system of patchy NPs composed of inorganic NP cores of gold triangular nanoprisms and polymer patches of polystyrene-*b*-polyacrylic acid (PS-*b*-PAA) to show polymer–polymer attraction can induce symmetry-breaking during grafting process. By tuning enthalpic interactions relative to entropic steric hindrance, we can introduce patch asymmetry beyond the conventionally considered features of core NPs such as their exposed facets[6,7] and local surface curvature[20,21]. Symmetry-breaking triangular prisms with a single-patch geometry are obtained with a yield of 82%. Quantitative agreements between experiment, theory, and simulation are observed at NP-level features such as the number of patches, as well as molecular characteristics like patch shape and size across both temperature and grafting density space. Our theory utilizes generalized scaling parameters for polymers such as the correlation size and the Flory-Huggins interaction parameter $\chi$ rather than their specific chemistry[25,26], enabling generalizability to other hybrid materials systems. Lastly, symmetry-breaking, single-patch prisms can assemble into nonconventional bowties, trimers, and tetramers connected by tip patches that exhibit intriguing plasmonic coupling in finite-difference time-domain (FDTD) electromagnetic calculations. The tip-to-tip distance of the assemblies is governed by interpenetration of "soft" polymer patches, which can also be quantitatively predicted by our theory. Combined, our experiment-modeling integration introduces a strategy of asymmetric grafting that can be utilized to design unconventional, symmetry-breaking patchy NPs exhibiting directional interactions and patch reconfigurability[17,27,28]. We additionally note that while previous studies have successfully predicted patch size for polymer grafted NPs[16,29], NP-level feature of patch placements and descriptors for their nanoscopic features remain elusive. As such, quantitative agreement between experiments and theory across multiple length scales presented in our work affords a high degree of precision that can be leveraged to design symmetry breaking polymeric grafting with nanometer-level precision. The fundamental understanding established in this work can guide patterning and engineering of solid surfaces using functional polymers for optical components[30], cellular engineering[31], sensors[32], data storage media[33], and organic devices[34], at nanometer precision.

## Results
### High-yield synthesis of single-patch gold triangular prisms
In our model system of gold triangular nanoprisms and PS-*b*-PAA (Fig. 1a, Methods), we show differentiated polymer grafting onto otherwise physiochemically and geometrically identical prism tips. Gold prisms (edge length: 62.5 ± 3.9 nm, thickness: 27.2 ± 3.7 nm, Supplementary Fig. 1) are first synthesized using the method in ref. 35. Next, an aqueous suspension of prisms of a controlled concentration is mixed with 2-naphthalenethiol (2-NAT) and PS-*b*-PAA in a dimethyl formamide/H₂O mixture (Supplementary Table 1) in a one-pot procedure to induce polymer patch formation. In doing so, nanoprisms attain hydrophobically-coated areas on all three tips via chemisorption of 2-NAT ligands[36] (Supplementary Fig. 2). The PS block of the

polymers can adsorb onto the ligand-coated tip area due to hydrophobic attraction, with the PAA block projecting outwards into the aqueous solution[17,18,37]. Here, we utilize the asymmetric grafting to overwrite the intrinsic three-fold symmetry of the nanoprisms during patch formation. We hypothesize that shifting the net chain–chain interaction from repulsive to attractive provides a direct handle to control the grafting symmetry (Fig. 1a). Repulsive chains seek to maximize the polymer configurational entropy, favoring sparse polymer distribution across all tips of the nanoprism. Conversely, strong van der Waals forces induce net attraction between chains, favoring chain localization to achieve singlet tip grafting.

We first validate the hypothesis of attraction-induced symmetry-breaking from the synthesis of single-patch prisms. We optimize the grafting temperature $T$ at 90 °C and a reaction time of 2 h in a one-pot mixture, followed by washing the NPs thoroughly to remove non-adsorbed polymers and redispersing the grafted NPs in water (Methods). The transmission electron microscopy (TEM) image in Fig. 1b clearly shows single-patch prisms (side view in Supplementary Fig. 3a). Reconstruction of a representative single-patch prism using TEM tomography (Fig. 1c, Supplementary Movie 1, Methods) reveals that the patch on the prism envelops only one tip. This 3D morphology is consistent with atomic force microscopy (AFM) images of the same sample (Supplementary Fig. 3c, Methods). The yield of single-patch prisms is 82%, while the percentage of the prisms with no recognizable patch (zero-patch) and with two patches (double-patch) are 9% for each (Fig. 1d). We note that no tri-patch prisms are observed in this condition. The chord length $l$ of the patches on the prism, the approximated radius $r_p$, and the maximum patch thickness $t_m$ of the patch (2D projected TEM view) all show narrow distributions (Fig. 1e inset, Supplementary Fig. 3b,d with $l = 16.8 \pm 1.8$ nm, $r_p = 14.6 \pm 2.3$ nm, and $t_m = 18.9 \pm 4.0$ nm, measured from 160 NPs). The positional offset of the patch center, $\triangle o$ (Fig. 1e), is small and monodisperse (2.0 ± 1.5 nm, 3.2% of the prism edge length), suggesting uniform positioning of the patches relative to the prism tip. These analyses indicate that polymer patches on the core NPs can be controlled with nanometer precision through the consecutive adsorption of individual polymer chains.

We note that the single-patch does not necessarily form on the prism tip exhibiting the highest surface curvature (Supplementary Fig. 4), differing from the curvature-driven polymer grafting reported in previous studies[20–22]. Control experiment of 12 min of reaction time (10% of the total reaction time, Methods), produces only small single-tipped patches, if any, on the NPs (Supplementary Fig. 5a). Such findings confirm that asymmetric polymer adsorption occurs from the reaction onset. In a sequential control experiment where the prisms are first incubated with 2-NAT and then incubated with polymers, the prisms out of incubation with 2-NAT-only can assemble into structures due to the hydrophobic attractions at all three tips (Supplementary Fig. 5b). Nevertheless, despite the 2-NAT decoration on all the three tips, the subsequent polymer grafting upon polymer addition generates single-patch prisms (Supplementary Fig. 5c), confirming that strong chain–chain interaction induces asymmetric grafting. Furthermore, the single-patch prisms prepared at 90 °C, when incubated at 110 °C (Supplementary 5d, e, Methods), become tri-patch prisms. This reversibility of patch formation is consistent with our hypothesis that the grafting symmetry is thermodynamically controlled by chain–chain interaction. Therefore, presented asymmetric polymer grafting strategy fundamentally differs from previous works using post-adsorption collapsing of uniformly grafted polymers[15–17], phase separation of ligands[18,38], curvature-induced grafting[20–22] and surface-pinned micelles[39].

### Experimental control of asymmetric polymer grafting
To examine the driving forces governing the symmetry breaking in patch formation, we study the underpinning chain–chain interaction as

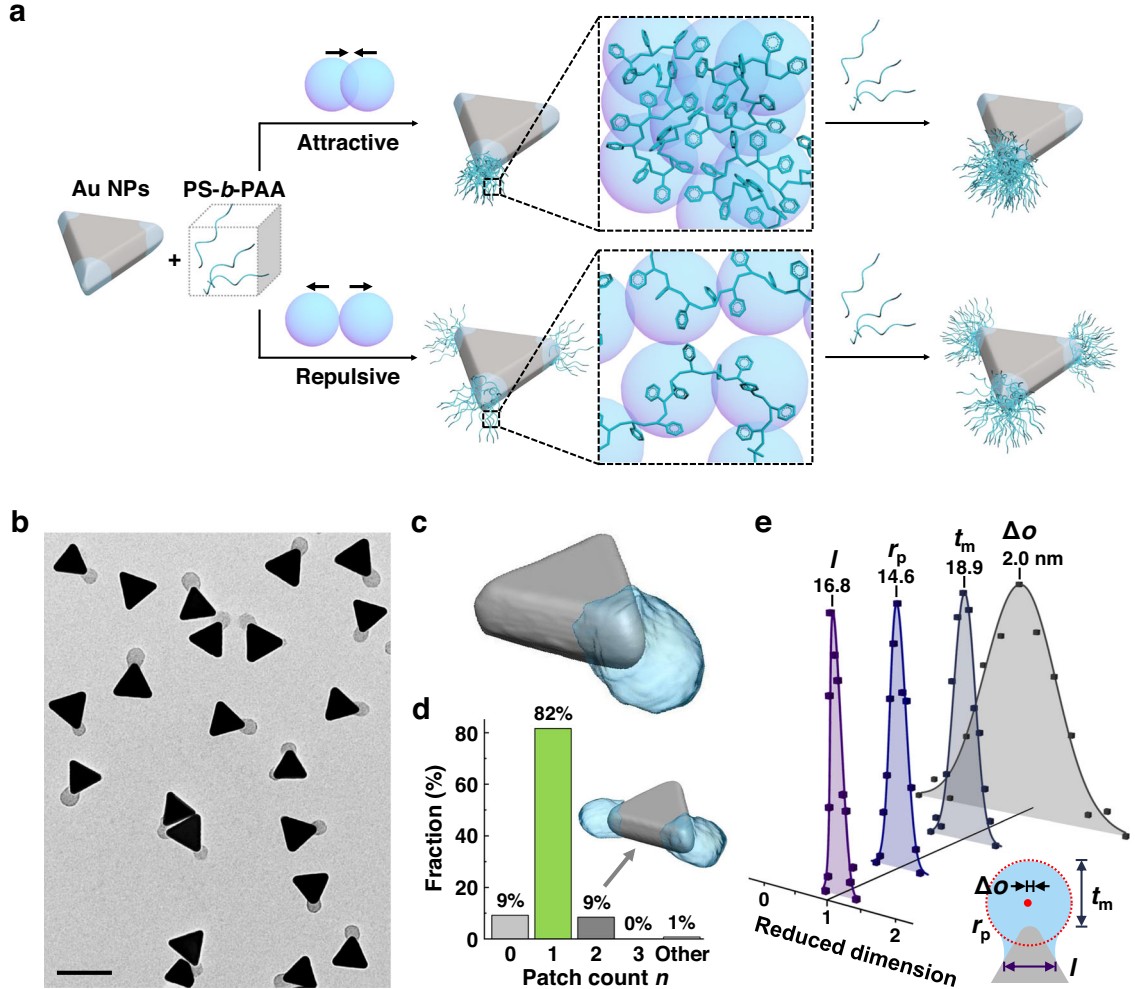

**Fig. 1 | High-yield synthesis of symmetry breaking single-patch gold triangular nanoprisms via asymmetric polymer grafting. a** Schematics of the asymmetric grafting by controlling the net chain–chain interaction. Net attraction leads to asymmetric single-patch prisms (top), whereas net repulsive interaction leads to symmetric tri-patch prisms (bottom). The prism tip surfaces are hydrophobic as illustrated by the light blue shadows. **b** Representative TEM image of single-patch gold prisms. **c** TEM tomography reconstruction of a single-patch prism. **d** Histogram of patch count per prism $n$, from the same sample presented in (**b**) (330 NPs were analyzed). The "Other" label corresponds to prisms with at least one prism edge fully coated. Inset: TEM tomography reconstruction of a double-patch prism. **e** Histograms of the patch shape parameters renormalized by the average value of each parameter: chord length $l$, fitted inscribed circle (red dotted circle) of the patch contour $r_p$, maximum patch thickness $t_m$, and offset of the patch center $\triangle o$ (noted as the red dot, defined as the centroid of the fitted inscribed circle of the patch contour) from the line connecting the prism center and the tip center. The averaged values (all with a unit of nm) are noted on top of the histogram. 160 NPs were measured. Scale bar: 100 nm.

a function of grafting temperature and effective local polymer concentration (Fig. 2a). At low grafting temperature $T$, chain–chain attraction drives strong localization, producing symmetry breaking patches (Fig. 2b). In addition to $T$, polymer–polymer attraction-driven asymmetric patch formation relies on a stochastic adsorption of the first few polymer chains onto one of three equivalent tips. To ensure symmetry breaking, local polymer concentration available to adsorb onto the NP tips needs to be sufficiently low. Otherwise, high local polymer concentration will disrupt the energetic gain afforded by chain–chain attraction between the first polymer attached to a NP tip and a free chain in solution, canceling out asymmetric graft localization. As more chains adsorb to the surface, thermodynamics dominates and chain–chain attraction provides the major driving force for asymmetric patch formation. We explicitly probe this concentration dependence by changing the 2-NAT concentration α (Fig. 2c). Higher α equates to more hydrophobic prism tips, attracting more polymer chains from the solution to the prism surface, thereby increasing the local polymer concentration. Conversely, a small α keeps the local polymer concentration low, enhancing asymmetric grafting effect.

Indeed, increasing grafting temperature $T$ leads to higher patch symmetry (reaction conditions in Supplementary Table 1). At a constant α of 50 nM, a temperature increase from 90 °C to 100 °C decreases the fractions of single- and double-patch prisms from 14 to 11%, and 67 to 25%, respectively (Fig. 2d, e, Supplementary Fig. 6a–c, 7a, Supplementary Table 2). Notably, increasing to 110 °C induces symmetric grafting with a 99% yield of tri-patch NPs, consistent with our previous report[36]. Raman spectra of the prisms show a constant peak intensity for the 2-NAT ligand[40] over the range of temperature (90–110 °C) (Supplementary Fig. 8a, Methods), suggesting negligible change in the 2-NAT ligand adsorption on NPs[41]. Therefore, the transition to tri-patch grafting at 110 °C is a result of weakened chain–chain attraction that produces equipartitioning of chains across all prism tips. This observation extends to the range of α from 25 nM to 75 nM that we use in this work where, at 110 °C, tri-patch prisms are consistently obtained (Supplementary Fig. 2a–d). These results emphasize systematic tunability of the grafting symmetry induced by strong chain–chain interaction to control the number of patches, despite the fact that all three tips of prisms are coated by 2-NAT. This is also

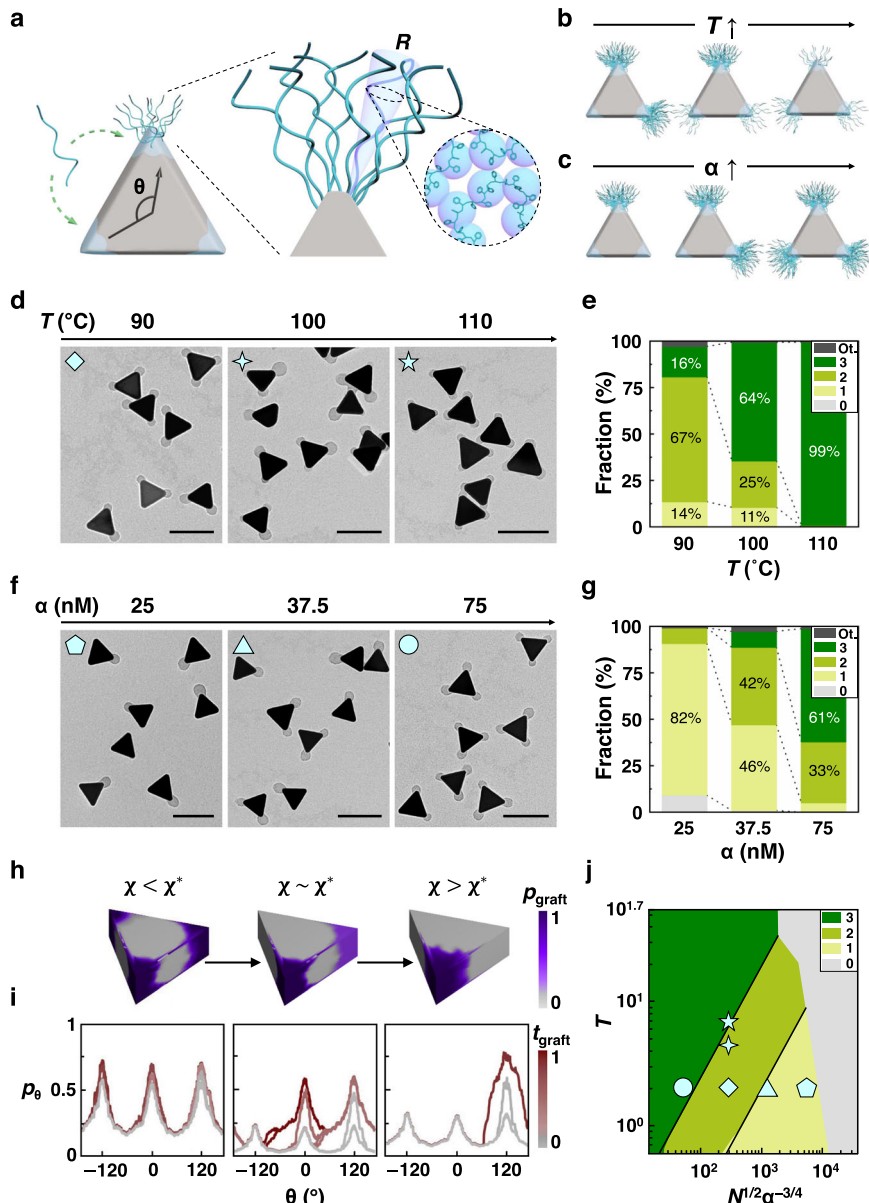

**Fig. 2 | Quantitative control of the asymmetric grafting. a** Schematic of the theoretical model of polymer chain grafting. Green dotted arrows (left) denote adsorption sites (tip area with the light blue shadows) on prisms for incoming polymer chains, either sites already occupied by other polymer chains or unoccupied. Zoomed-in view (middle) shows the theoretical model of grafted polymer chains (end-to-end distance $R$) on the tip of prisms, as a function of surface position $\Omega$. **b, c** Schematics showing the effect of increasing (**b**) temperature $T$ and (**c**) ligand concentration $\alpha$, the ratio of 2-NAT molar concentration to the optical density of the prism suspension at its maximum extinction wavelength. **d, f** TEM images of patchy prisms at (**d**) increasing $T$ ($\alpha = 50$ nM, fixed) and (**f**) increasing $\alpha$ ($T = 90$ °C, fixed). **e, g** Histograms of NPs with the patch count $n$ synthesized from the corresponding conditions presented in (**d**) and (**f**), respectively. Counted from 150–220 NPs per condition. Shading colored to $n$ as noted in the legend. 'Ot.' in legend refers

to the prisms with at least one prism edge coated with polymer patch. **h** Maps of 3D grafting probability $p_{graft}$ at varied $\chi$ values. **i** Plots of patch profile $p_\theta$ during the grafting simulation time $t_{graft}$. The $p_\theta$ is obtained by projecting $p_{graft}$ over 2D projected prism contour and integrating it as a function of angle $\theta$ (defined in (**a**)). We define the grafting "time" $t_{graft}$ as the ratio of current grafting density to the targeted grafting density of the simulation. **j** Phase diagram of patch count $n$ (shading colored to $n$ as noted in the legend) across the parameter space: grafting temperature $T$, chain length $N$, and ligand concentration $\alpha$. The phase diagram is partitioned to different colored regions following simulation results. The black lines are phase boundaries predicted by theory. The cyan-colored open symbols in the phase diagram correspond to the experimental results labeled in (**d**) and (**f**). Scale bars: 100 nm.

consistent with our polymer scaling theory prediction that—as compared to the polymer chains of molecular weight 20,000 g/mol —short ligands of 160 g/mol (i.e., 2-NAT) are not necessarily preferentially accumulated on one tip.

Next, we consider the effect of local polymer concentration by setting the 2-NAT concentration $\alpha$ at 25, 37.5, and 75 nM for a constant $T$ of 90 °C. Overall, TEM images show an increase in patch count per prism $n$ from $0.99 \pm 0.42$ to $2.54 \pm 0.59$ (Fig. 2f, Supplementary

Figs. 6d–f, 7b). Corresponding Raman spectra show increased 2-NAT adsorption, implying that more polymer chains can be attracted to the prism surface (Supplementary Fig. 8b). The fraction of single-patch prisms decreased from 82 to 5% with increased $\alpha$ (Fig. 2g, Supplementary Table 3), and the fraction of tri-patch prisms increased from 0 to 61%. Both trends are consistent with our expectation that increasing $\alpha$ favors symmetric grafting. At $\alpha = 50$ nM, the fraction of double-patch prisms is 69%. These results emphasize systematic tunability of

asymmetric grafting strategy to control the number of patches and symmetry of patchy prisms.

## Mechanism of symmetry-breaking in patch formation

To quantitatively model the polymer–polymer attraction-induced asymmetric grafting, we adapt star polymer scaling theory[42] to explicitly incorporate entropic and enthalpic energies arising from chain localization on prism surfaces (Fig. 2a, Methods, Supplementary Note 1). Previous works operated in the limit where polymers maximize interactions with solvents[20,21]. We build on these developed extensions to incorporate chain–chain attractions by introducing a non-zero Flory-Huggins parameter[43] $\chi$, with temperature dependency[25,26] $\chi$-$T^{-1}$. Varying $\alpha$ controls the chain concentration locally available at the prism tip and is related to the grafting density[23] $\sigma$. The end-to-end distance $R$ of an end-attached polymer chain (i.e. patch size, Supplementary Note 2) as a function of surface position $\Omega$ is derived to be:

$$R \sim r_o \alpha^{1/5} v_o^{1/5} [1 - 2\chi]^{1/5} b_l^{2/5} \left[ \frac{N}{\Omega} \frac{b_l}{r_o} \right]^{3/5} \quad (1)$$

where $r_o$ is the NP in-sphere radius, $v_o$ is the excluded volume of a Kuhn monomer of size $b_l$, and $N$ is the number of Kuhn monomers in a chain. Phenomenologically, chain–chain attraction increases with increasing $\chi$. Above a critical value $\chi^*$, chain–chain attractions are so strong that they win over competing enthalpic gains associated with chain–solvent interaction as well as entropic penalties arising from local chain confinement, resulting in asymmetric patch grafting. This crossover occurs at $\chi^*$ where

$$\chi^* \sim 2^{-1} \left[ 1 - r_o^{1/2} \alpha^{1/4} \Omega^{-3/4} v_o^{-1} N^{-1/2} \right] \quad (2)$$

The theory predicts the experimentally observed symmetry breaking as a function of $T$ and $\alpha$ (Fig. 2j).

To visualize how symmetry-breaking occurs, we employ Monte Carlo (MC) grafting simulations[44,45]. Here, we quantify the grafting probability of a chain $p_{graft}$ at different surface positions $\Omega$ as a function of $\chi$ and $\sigma$ (Supplementary Note 3, Supplementary Movie 2). In the high temperature regime ($T = 110$ °C, $\chi < \chi^*$), the final $p_{graft}$ is symmetric across all three tips (Fig. 2h, left). Conversely, at $\chi > \chi^*$, single-patch prisms appear as the $p_{graft}$ shifts to favor only one tip (Fig. 2h, right). Evolution of the patch profile $p_\theta$ is plotted by projecting $p_{graft}$ over the prism contour and integrating it as a function of angle $\theta$ (labeled in Fig. 2a). When chain–chain attraction is strong enough to overcome the confinement energy cost ($\chi > \chi^*$), the initially grafted chains actively recruit other chains to the same location, iteratively augmenting $p_\theta$ on only one of the prism tips (Fig. 2i, right). We note that a second prism tip can become decorated with polymer chains during the grafting process. But, because grafting is a stochastic process, those chains can subsequently detach back into solution and continue to sample the prism surface to find the most favorable site–the tip already occupied by other chains–to maximize the polymer–polymer attraction of the whole system, as also guided by their long interaction range on the tips (Supplementary Note 4 and Supplementary Fig. 9). This equilibration process is fast, as modeled by an obvious preferential adsorption kinetics in our simulation (Fig. 2h–i). In contrast, at $\chi < \chi^*$, all three tips of a prism are concurrently occupied from the onset of grafting (Fig. 2i, left). When the two energetic contributions are on par with each other, the system can freely choose to graft to one of the two remaining tips ($\chi$-$\chi^*$, Fig. 2h, i, middle). At first, only the tip at $\theta = 120°$ is grafted. Subsequent addition of more chains at the same location progressively increases the confinement cost, making it less energetically efficient to graft to the same tip. Eventually, the system shifts to favor grafting to another tip ($\theta = 0°$) to create a double-patch prism. Consistent with our experiment, the $p_{graft}$ for changing ligand concentration shows that increasing $\alpha$ induces equivalent grafting on

every tip (Supplementary Figs. 10 and 11, Supplementary Movie 2). These results verify our hypothesis that shifting the net chain–chain interaction from repulsive to attractive drives asymmetric grafting.

We performed simulations beyond the experimentally sampled parameter space to provide a complete composite phase diagram for symmetry-breaking patchy NPs. The number of patches $n$ is plotted as a function of $T$, $\alpha$, and $N$, whose phase boundaries collapse into a master curve at reduced parameter dimensions ($T$-$N^{1/2}\alpha^{-3/4}$, Fig. 2j, Supplementary Note 3). Theoretical transitions (black line, Fig. 2j) in the number of patches occur when chain confinement costs exceed the energetic gains from favorable chain–chain interactions and the system opts to create a new patch at an open tip. Experimental observations within this phase diagram are denoted as open symbols in Fig. 2j, all falling into the correct regimes of $n$. As shown in Fig. 2d and Supplementary Fig. 2a–d, tri-patch prisms are consistently obtained at 110 °C regardless of $\alpha$, while asymmetrically grafted single-patch (Fig. 1) and two-patch prisms (Fig. 2d) are formed at 90 °C. Similarly, in our phase diagram (Fig. 2j), increasing temperature along the y-axis alone can control the patch count across a range of $\alpha$, emphasizing the grafting temperature as an effective knob by controlling chain–chain attraction, and thus the patch grafting symmetry. This agreement suggests that the composite phase diagram can aid in leveraging the interplay among $T$, $\alpha$, and $N$ to direct asymmetric grafting on NPs.

We verify the generalizability of the strong chain–chain interaction induced asymmetric grafting using differently shaped NPs and polymer chains of different lengths. On one hand, octahedron and bipyramid having one to three patches are formed, despite their number of vertices (i.e., 6 and 7, respectively, Supplementary Fig. 12). On the other hand, using a slightly longer polymer $PS_{230}$-$b$-$PAA_{49}$, we see similar results as for $PS_{154}$-$b$-$PAA_{51}$. At a low grafting temperature of 90 °C, single-patch prisms are obtained whereas increasing temperature to 110 °C retrieves tri-patch prisms (Supplementary Fig. 13a, b). Interestingly, by using even longer polymer chains $PS_{394}$-$b$-$PAA_{58}$, we can get single-patch prisms even at 110 °C (Supplementary Fig. 13c, d). It is also consistent with the phase diagram (Fig. 2i) as one move towards the right along the x-axis, implying that longer chains enhance asymmetric grafting effect.

## Nanometer precision control of patch size and shape from both experiment and scaling theory

Asymmetric grafting also controls the nanoscopic features of size and shape of each individual patch, which are quantitatively captured by our theory. In Fig. 3a, we color patches based on local thickness $t$ in the TEM images of patchy prisms synthesized at different experimental conditions to highlight patch size dependence on $T$ and $\alpha$ (Supplementary Note 5). Overall, patches become smaller as $T$ increases from 90 °C to 100 °C and 110 °C (ii, iv, v in Fig. 3a, $\alpha = 50$ nM), and become bulkier as $\alpha$ decreases from 75 to 50 nM and 25 nM (i, ii, iii in Fig. 3a, $T = 90$ °C). These observations match with the MC simulation results (Fig. 3b, Supplementary Note 3), where patches are rendered as envelopes of grafted polymer chains. Increasing $T$ weakens chain–chain attraction and reduces the number of chains per tip, producing thinner patches. Interestingly, at 100 °C, representative tri-patch prisms have both types of patches congruent in size to those obtained from 90 °C and at 110 °C within the same NPs (Supplementary Fig. 14a), which is indicative of a thermodynamic coexistence of patch morphology at 100 °C, consistent with the symmetry-breaking threshold predicted from our theory ($T = 103$–106 °C). On the other hand, decreasing $\alpha$ produces thicker patches (Fig. 3a, Supplementary Fig. 14b). This is a direct result of increased chain localization to certain tips, where increasing confinement extends the chain's end-to-end distance $R$ and ultimately manifests as larger patches.

Molecular-level insights can also provide detailed quantification of patch area $A$ (2D projected, Supplementary Note 5) and its

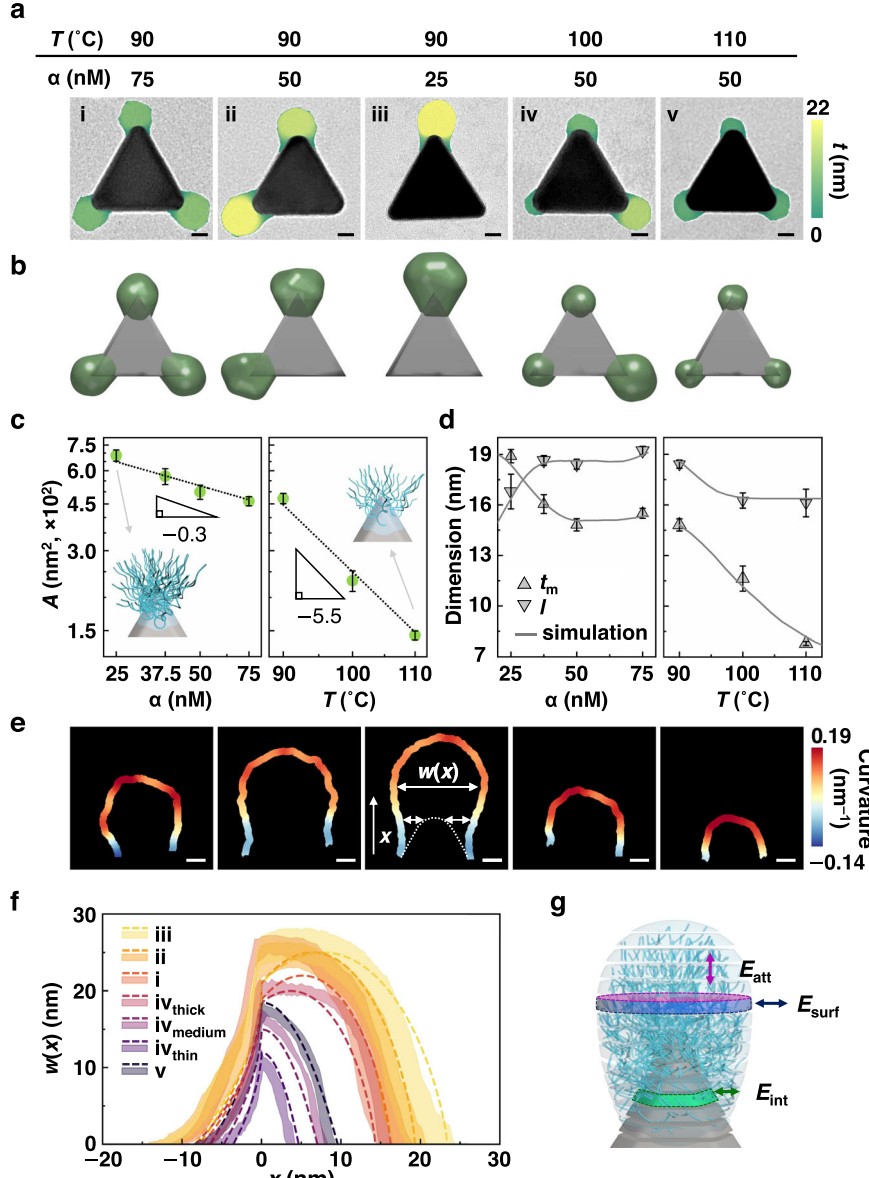

**Fig. 3 | Nanometer precision in size and shape of patches from asymmetric grafting. a** Representative TEM images of patchy prisms. The patches are color-coded to their local thickness $t$. **b** Representative simulated patchy prisms obtained at the conditions corresponding to (**a**). **c** Plots showing patch area $A$ (in 2D projected TEM view) decreases with increasing $\alpha$ (left, at fixed $T = 90\,°C$) and $T$ (right, at fixed $\alpha = 50\,nM$) in log–log scale with power law fitting. Mean values are shown and error bars represent the standard deviation. **d** Plots of $l$ and $t_m$ as a function of (left) $\alpha$ and (right) $T$ from experiments (symbols: mean values and standard deviation) overlaid with predictions from simulation (solid lines). **e** The contour of the topmost patch in (**a**), color-coded by local surface curvature. Local patch width $w(x)$ is defined as a function of $x$, an axis vector connecting from the center to each tip of a

prism. The position of the tip refers to $x = 0$. Note that $w(x)$ at $x < 0$ region was measured by subtracting the prism width $p(x)$ from the width of patch contour at $x$. **f** Plot of $w(x)$ (the standard deviation of 10 representative $w(x)$ as colored shadow for the prisms synthesized at the reaction conditions listed in (**a**) overlaid with the corresponding predictions from theory (dotted lines). Note that condition iv in (**a**) has a coexisting range of flattened and bulky patch shapes categorized into thin, (iv$_{thin}$), medium (iv$_{medium}$), and thick (iv$_{thick}$). **g** Schematic showing the pendant-like shape of large patches governed by the balance of three interactions: interaction between polymer and 2-NAT-coated prism tip ($E_{int}$), attraction between all monomers presented within the layer ($E_{att}$), and surface tension cost from contact with the surrounding solvent ($E_{surf}$). Scale bars: 10 nm (**a**), 5 nm (**e**).

geometric components as a function of $T$ and $\alpha$. The patch area $A$ can be described as $A \sim \delta RD$, where $\delta$ is the fractional coverage of polymer chains over the adsorption sites and $D$ is the corresponding interchain separation. Plugging scaling relationships we derived gives $A \sim \alpha^{-3/10} T^{-11/2}$ (Supplementary Note 6), consistent with that from the experiment ($\sim \alpha^{-0.36}$ and $\sim T^{-5.9}$, Fig. 3c, Supplementary Fig. 14c–e). Decomposing patch area into two contributing geometric components – chord length $l$ and maximum patch thickness $t_m$ – also shows excellent quantitative agreement between experiment (Fig. 3d, symbols) and simulation (Fig. 3d,

solid lines). Notably, $t_m$ follows a similar dependence to patch area $A$ on $T$ and $\alpha$. Chord length $l$ increases with increasing $\alpha$ from 25 nM to 75 nM at a constant $T$ of 90 °C, reflecting increased adsorption sites on the tip (Fig. 3d, left, Supplementary Figs. 8b, 14f). Furthermore, $l$ decreases (12.5%) as $T$ increases from 90 °C to 110 °C (Fig. 3d, right) due to a reduced number of chains localized at the tip. Since our scaling theory is agnostic of chemistry level details in its derivation, direct comparison with experiments requires a scaling constant to match to a specific experimental condition. Doing so allows us to relegate molecular details such as specific

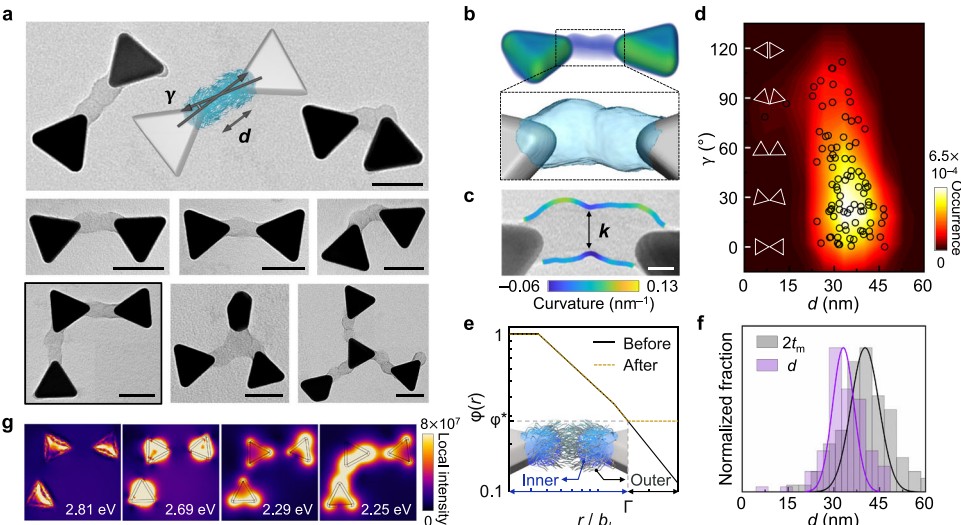

**Fig. 4 | Self-limited assembly of single-patch prisms by patch–patch inter-penetration. a** TEM images of nano-bowties, trimers, and tetramers exhibiting tip-to-tip connection. Schematics of a nano-bowtie labeled with interparticle angle γ and tip-to-tip distance $d$ (inset). **b** TEM tomograph of a nano-bowtie and 3D reconstruction of merged patches. **c** Curvature analysis of merged patches. **d** Heatmap of γ and $d$ by the kernel density estimation overlaid with the scatter plot obtained from experiment: γ = 39.9 ± 30.4°, $d$ = 33.6 ± 6.7 nm. **e** Theoretically calculated monomer density profiles φ($r$) before (black line) and after merging (black line with a green shadow) as a function of the distance away from a tip $r$ normalized by Kuhn monomer size $b_l$ at interparticle angle γ of 0°. Inset: schematics of the theoretical model of patch–patch interpenetration. The crossover distance Γ divides a patch into inner and outer regions, where intrapatch and interpatch chain–chain interactions are dominant, respectively. φ* refers to the equilibrium monomer concentration after merging. **f** Normalized histograms of $d$ (purple bars) and doubled patch thickness $2t_m$ (gray bars) measured from experimental TEM images and from simulations (solid lines of corresponding color). **g** Plasmonic near-field LSPR maps calculated by solving Maxwell's equations using Finite Element Modeling (FEM) of the trimer boxed in (**a**) at excitation energies corresponding to the modes (2.81 eV: face; 2.69 eV: edge; 2.29 eV: tip antibonding; 2.25 eV: tip bonding). The orientation of the electric field is averaged in-plane, with incident excitation impinging on the particles from perpendicular to the plane. Scale bars: 50 nm (**a**), 10 nm (**c**).

polymer types, surface chemistry, and/or side group interactions into a scaling prefactor[26].

The asymmetric grafting induced by polymer–polymer attraction also determines nanoscopic patch shape, such as "pendant-like" large patches and "dome-like" small patches. As shown in Fig. 3e, large patches (i, ii, and iii, Fig. 3a) have regions of negative and zero local curvature near the prism tip (Supplementary Note 5, Supplementary Fig. 14g). Quantitatively, a patch width $w(x)$ (defined in Fig. 3e) is measured to describe patch shape as a function of the distance from the prism tip, $x$, along the direction connecting the center of a prism to that of the tip (Fig. 3f, Supplementary Note 5). We model patch shape by discretizing patches into layers going away from the prism tip (Fig. 3g, Supplementary Note 6), which agrees with experimental profiles. The contact angle of each layer relative to the previous layer is determined using the Young-Laplace surface wetting equation[46], balancing three interactions: attraction between polymers and prism tip ($E_{int}$), attraction between all monomers present within the layer ($E_{att}$), and surface tension cost from contact with the surrounding solvent ($E_{surf}$). Pendant-like patches are predicted when chain–chain interaction wins over surface tension cost (Fig. 3f, dotted lines i–iii). This adds more chains to the patch, inducing localized chain stretching that then drives patch elongation (Supplementary Note 6). Overall, the "shark-fin" shape of $w(x)$ reflects the different emergent polymer regimes of the patch. Despite differences in the shape of the $w(x)$ profiles, all curves collapse onto each other at the regimes near the surface of the prisms ($x < 0$ region, in Fig. 3f). We attribute the collapse of the $w(x)$ curves to chain crowding near the prism surface, where the monomer density profile is uniform[47], leading to similar chain–chain interaction and surface tension cost across different conditions. Following the concentrated regimes, chains enter a semi-dilute limit where they do not feel the effect of solvent penetration, producing a more constant patch width. With increasing distance from the surface, the number of monomers contributing to the patch decreases more and more, giving rise to a decreasing patch width.

## Colloidal bowties and clusters formed from patch merging of single-patch nanoprisms

The polymer patches on prisms are "soft" and, at locally high particle concentration, chains interact not only with others within the same patch, but also with those in patches on neighboring NPs. As a result, patches can interpenetrate each other, serving as reconfigurable hinges linking NPs into a tip-to-tip arrangement. We observe dimerization of single-patch prisms to form bowties (Fig. 4a, Supplementary Fig. 16a–d). 3D morphology of the dimers obtained by TEM tomography reveals that the patches are smoothly merged (Fig. 4b, Methods, Supplementary Movie 3). The contour of the merged interface shows a flat region with a neck width $k$ of 16.5 ± 2.9 nm (Fig. 4c, Supplementary Note 5, Supplementary Fig. 16e, f), suggesting chain rearrangements within patches to maximize chain–chain contacts. Statistical analysis of dimer configurations shows a wide distribution of interparticle angle γ and narrow distributions of the tip-to-tip distance $d$ and the projected merged patch area $A_m$ (Fig. 4d, Supplementary Fig. 16g, and Supplementary Note 5).

To understand the flexible reconfiguration of the patches during patch merging, we expand our theory from the dilute to semi-dilute regime, above which patch–patch interactions occur (Supplementary Note 8). Figure 4e shows the monomer concentration profile φ($r$) of the patches before and after merging as a function of the distance away from one tip $r$. Before merging, φ($r$) monotonically decreases to zero with increasing $r$. After merging, the grafted chains split into inner and outer regions (Fig. 4e, inset) demarcated by the crossover distance Γ. In the outer region ($r/b_l > Γ$), chains are no longer influenced by the NP core but only by local chains from either the same patch or the patch from the interacting prism. As a result, they behave analogously to traditional polymer solutions[42] and reorganize to attain a uniform distribution in the space between patches. Doing so minimizes contribution to the free energy from the monomers occupying the space between interacting prisms that subsequently serves to stabilize patch merging[42]. Using Γ and the post-merged equilibrium monomer

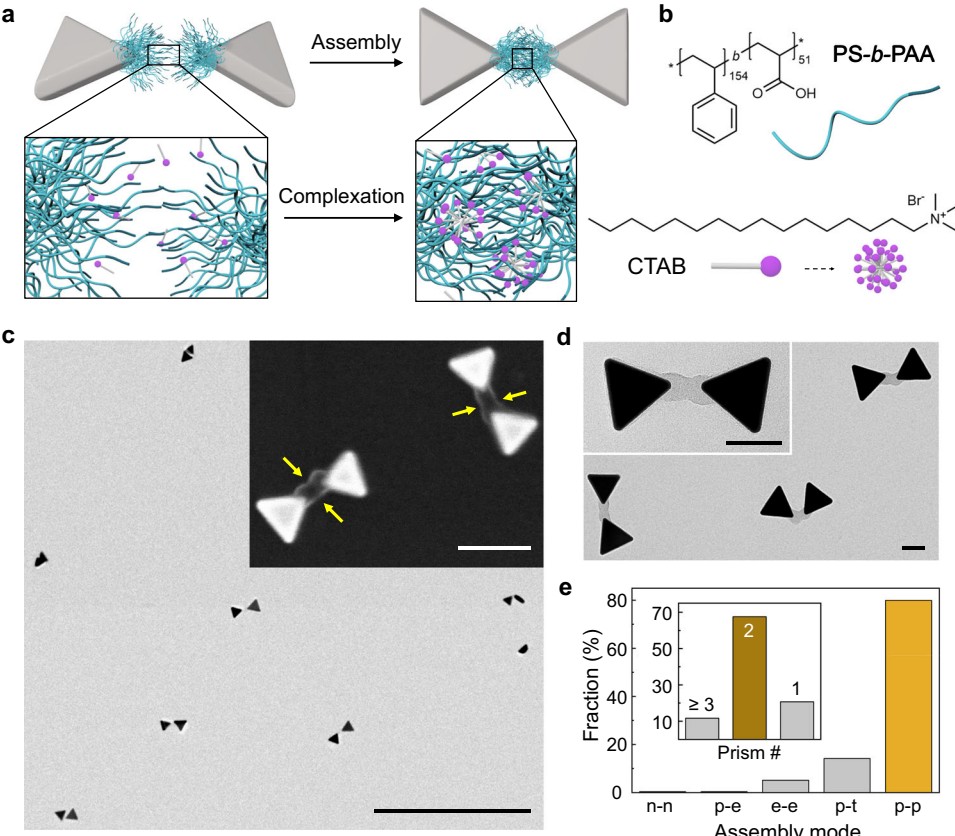

**Fig. 5 | Patch-to-patch selective assembly of single-patch prisms. a** Schematics of the CTAB-induced patch complexation mechanism. **b** Chemical structure of PS-*b*-PAA and CTAB involved in patch-patch selective assembly. **c** Low magnification TEM and scanning electron microscopy (SEM) (inset) images of dimers. Yellow arrows highlight the merged patch interface. **d** High magnification TEM images of patch-patch assembled dimers. **e** Yields of assembly motifs in dimers. The notations in the *x*-axis mean as follows: n-n: between non-patched prism tips. p-e: patch to prism edge. e-e: between prism edges. p-t: patch and non-patched tip. p-p: between patches. A total of 376 structures are counted. Inset: the yield of dimers compared to assemblies of trimers or larger (noted as '≥ 3') and nonassembled NPs (noted as '1'). Scale bars: 1 μm, 100 nm (inset) (**c**), 50 nm (**d**).

concentration φ*, the optimal patch interpenetration ratio λ is predicted to be 0.164 (where λ = d/2t_m) and γ is 37.2 ± 38.7° (Fig. 4f, Supplementary Note 8, and Supplementary Fig. 16h), matching well with experiment. This merging driving force provides compelling evidence for patches as flexible hinges for designing next-generation mechanical metamaterials[48].

We investigate the emergent optical properties of tip-to-tip assembly thorough flexible patch "hinges". Previous studies showed nano-bowties have 'hot spots,' where the electromagnetic fields are orders of magnitude more intense than the surroundings due to the interparticle coupling[49,50]. In contrast to those, the assemblies via patch-merging have distinct spacing *d*, possess wide angular distribution γ, and are extendable into larger structures using double- and tri-patch prisms as "linkers" and single-patch prisms as "terminators"[51,52]. As proof of concept, we demonstrate dimers, trimers and tetramers (Fig. 4a) and show the map of near-field intensity arising from their localized surface plasmon resonance (LSPR) as having asymmetric modes[53] (Supplementary Note 9, Supplementary Figs. 17 and 18). The trimer (boxed in Fig. 4a) shows both symmetric bonding (2.25 eV) and anti-symmetric antibonding (2.29 eV) modes. Such decomposition of the tip modes also appears in other linear and branched structures (Supplementary Fig. 18) exhibiting emergent plasmonic properties applicable to security, sensing devices[54], and spectral fibers[55].

To further increase the yield of selective, patch-to-patch assembly that enhances plasmonic coupling, we develop an assembly method based on surfactant-induced complexation, which has been employed for polymer film formation[56] and nucleotide purification[57] but not for directing NP assembly. According to previous studies[58,59], cationic surfactants (e.g., CTAB) facilitate aggregation of anionic polyelectrolytes (e.g., PAA) by forming a surfactant-polyelectrolyte complex (Fig. 5a, b). The head group of CTAB can attach onto the polyelectrolytes by electrostatic attraction, followed by hydrophobic attraction among the tails driving the aggregation of polymers to which those tails are attached. Using this method for our single-patch prisms, we obtain 68% dimeric assemblies (376 structures counted, Fig. 5c–e, Supplementary Fig. 19) and 21% unassembled individual prisms. Importantly, among the dimeric assemblies, over 80% are bowties attached by the tip patches. Thus, this result shows greatly increased yield compared to the random patch merging shown in Fig. 4, implying patch-patch selective, self-limited assembly. Additionally, when we use NaCl solution (concentration varied from 22.2 mM to 85.7 mM) and simply screen the charge-charge repulsion among patches, we do not obtain patch-to-patch assembly. Instead, the single-patch prisms are assembled through attachment of their non-coated gold surfaces (via edges or facets) as governed by strong van der Waals attraction between the gold (Supplementary Fig. 20), further highlighting the critical role of surfactant in favoring selective patch-patch assembly. When we apply the same surfactant-induced complexion method to tri-patch prisms, we observe consistently tip-to-tip connected networks (Supplementary Fig. 21).

We introduced a one-pot strategy to synthesize symmetry-broken patchy NPs by utilizing the polymer–polymer attraction-induced asymmetric grafting, where grafted polymers on prism tips actively

recruit free chains onto nearby adsorption sites, overwriting the intrinsic physicochemical symmetry of NPs. Detailed size and shape comparison of the patches in experiment and theory demonstrates that selective molecular adsorption can be achieved at nanometer precision by leveraging asymmetric grafting effect. The conceptual platform we provide can be generalized to other NPs and polymers. For example, extension of the asymmetric patch grafting to more complex NPs (e.g., cubes, octahedra, etc.) can potentially generate a library of symmetry-broken patchy NPs for self-propelling and spinning active nanomaterials with shape and symmetry-dependent locomotive trajectories[60]. The experiment-theory connection established for the polymeric patches can guide nanometer-scale precision in patch design, which can be extended to the predictions of "soft" patch behaviors such as shape reconfiguration[17,27], adaptation to stimuli[28,61], and directed assembly of flexible structures[22]. The essence of asymmetric grafting is to overwrite the intrinsic geometry and local curvature of the grafting surface by controlling chain–chain interactions. Therefore, asymmetric grafting provides a way to imprint complicated molecular patterns on surfaces for fabrications of functional modules for optical components[30], cellular engineering[31], sensors[32], data storage media[33], and organic devices[34].

## Methods

### Chemicals

Gold(III) chloride trihydrate (≥99.9% trace metals basis, HAuCl$_4$·3H$_2$O, Sigma-Aldrich), cetyltrimethylammonium chloride (CTAC) (>95%, C$_{19}$H$_{42}$ClN, TCI), sodium iodide (99.999%, NaI, Sigma-Aldrich), L-ascorbic acid (BioXtra, ≥99.0%, Sigma-Aldrich), sodium hydroxide (99.99%, NaOH, Sigma-Aldrich), cetyltrimethylammonium bromide (CTAB) (BioXtra, ≥99.0%, C$_{19}$H$_{42}$BrN, Sigma-Aldrich), silver nitrate (≥99.0%, AgNO$_3$, Sigma-Aldrich), potassium bromide (99.999%, KBr, Acros), hexadecylpyridinium chloride monohydrate (CPC) (>98.0%, C$_{21}$H$_{38}$ClN·H$_2$O, TCI), hydrochloric acid (99.999%, HCl, Alfa Aesar), benzyldimethylhexadecylammonium chloride (BDAC) (cationic detergent, Sigma-Aldrich), polystyrene-block-poly(acrylic acid) (PS-*b*-PAA) (PS$_{154}$-*b*-PAA$_{51}$, M$_n$ = 16,000 for the PS block and M$_n$ = 3700 for the PAA block, M$_w$/M$_n$ = 1.04; PS$_{230}$-*b*-PAA$_{49}$, M$_n$ = 24,000 for the PS block and M$_n$ = 3500 for the PAA block, M$_w$/M$_n$ = 1.02; PS$_{394}$-*b*-PAA$_{58}$, M$_n$ = 41,000 for the PS block and M$_n$ = 4200 for the PAA block, M$_w$/M$_n$ = 1.08, Polymer Source Inc.), 2-naphthalenethiol (2-NAT, 99%, Sigma-Aldrich), N,N-dimethylformamide (DMF) (anhydrous, 99.8%, Sigma-Aldrich), acetone (≥99.5%, Fisher Chemical), isopropanol (99.9%, Fisher Chemical), and Spin-X centrifuge tube filters (cellulose acetate membrane, pore size 0.22 μm, non-sterile, Corning Inc.) were purchased and used without further purification. Nanopure water (18.2 MΩ·cm at 25 °C) purified by a Milli-Q Advantage A10 system is used throughout the work.

### Synthesis and purification of gold nanoprisms (stock solution I)

The gold nanoprisms were synthesized following a literature method[35,36], with a slight modification for a 10 fold scale-up synthesis. Specifically, freshly prepared aqueous solutions of CTAC (16 mL, 100 mM), HAuCl$_4$ (800 μL, 25.4 mM), NaI (1.5 mL, 10 mM), NaOH (200 μL, 100 mM), and ascorbic acid (800 μL, 64 mM) were sequentially added to 80 mL water in a 125 mL Erlenmeyer flask with a 40 × 8 mm magnetic stir bar. The solution was stirred at 200 rpm for 10 s after each addition. After the addition of ascorbic acid, the solution was stirred at 500 rpm for 30 s, where the reaction mixture rapidly changed from light yellow to colorless, indicating the reduction of Au$^{3+}$ to Au$^+$. The aqueous solution of NaOH (550 μL, 100 mM) was added again into the flask to reduce Au$^+$ to Au$^0$. The reaction mixture was stirred at 200 rpm for 2 min, followed by left undisturbed for 8 min, where the solution color turned into dark blue, implying the growth of prisms. The reaction mixture was divided equally and transferred to three 50 mL centrifuge tubes (Corning Inc.), followed by centrifuging

twice (5850 x g for 10 min each) to remove the unreacted reagents. After the first round of centrifugation, 33.18 mL of the supernatant was removed from each tube, and the sediment (100 μL) was re-dispersed by adding 29.95 mL water into each centrifuge tube. After the second round of centrifugation, 30 mL of the supernatant was removed, and the sediment (50 μL) containing gold prisms was re-dispersed by adding 3.55 mL water into each centrifuge tube, then all three of them combined into one 15 mL centrifuge tube to make final 10.8 mL of prism solution. Note that all the glassware used in this synthesis are washed with aqua regia (mixture of HCl and HNO$_3$ with a volume ratio of 3:1), fully rinsed with water, and dried before use.

The gold prisms underwent three rounds of purification process to remove small gold polyhedral particles and large plate-like impurities by depletion attraction induced aggregation in a concentrated CTAC solution, following our previous literature method with a slight modification[36]. Specifically, the as-synthesized prism solution (10.8 mL) was mixed with an aqueous solution of CTAC (2.134 mL, 1 M) in a 15 mL centrifuge tube (a final CTAC concentration of 165 mM). The solution was left undisturbed overnight so that the prisms and large plate-like impurities stack into columns by depletion attraction and sediment as black solids, while the small gold polyhedral particles stayed dispersed in the top solution (reddish purple color). Next 12.924 mL of the top solution was removed from the centrifuge tube and the remaining 10 μL sediment was re-dispersed with 10.79 mL water. The solution went through the second round of purification by repeating the same procedure as in first round to further remove the small impurities, followed by 10 μL sediments re-dispersed with 10 mL water. The solution went through the final round of purification to remove the large plate-like impurities in a new 15 mL centrifuge tube by adding CTAC solution (1.11 mL, 1 M) into the tube (a final CTAC concentration of 100 mM). The solution was left undisturbed overnight, and 11 mL of the dark bluish green top solution containing the purified prisms was collected in a new 50 mL centrifuge tube. The remaining 120 μL sediment contains the large plate-like impurities that more easily aggregate and sediment. Successful purification was indicated by a single peak of the UV-Vis spectrum at the maximum extinction wavelength ($\lambda_{max}$) around 645 nm (Supplementary Fig. 1a). The collected prism solution was then diluted with 24 mL water and centrifuged at 4500 x g for 15 min. After centrifugation, 34.98 mL of the supernatant was removed. The 20 μL sediment was re-dispersed with 4 mL water, mixed with an aqueous solution of CTAB (1 mL, 100 mM), and reached a final CTAB concentration of 20 mM (stock solution I). A typical stock solution I has ~ 7.5 optical density (OD) at $\lambda_{max}$, which was sufficient for the following preparation and characterization of patchy nanoprisms.

### Preparation of prism solution of a desired concentration (stock solution II)

The single-patch nanoprisms (Fig. 1b) were synthesized based on the procedure in our previous report[36], with a modification in reaction temperature and reagent concentration. In a typical experiment, stock solution I was freshly centrifuged two times to reduce the CTAB concentration, and precisely diluted to 5 OD at $\lambda_{max}$ (stock solution II). Specifically, stock solution I (0.5 mL) was diluted with 1 mL water in a 1.5 mL microcentrifuge tube, followed by centrifugation at 4200 x g for 15 min. After centrifugation, 1.4 mL of the supernatant was removed from the tube. The sediment (100 μL) was re-dispersed with 1.4 mL water in a 1.5 mL microcentrifuge tube (CTAB concentration of 0.44 mM). The solution was centrifuged at 3100 x g for 15 min. After the centrifugation, 1.488 mL of the supernatant was removed and the remaining 12 μL sediment was diluted with 400 μL of water. After measuring UV-Vis spectrum of the diluted prism solution (30 μL), the remaining solution of 382 μL was further diluted with water (~ 295 μL), depends on the dilution factor calculated from the spectrum to precisely make a stock solution II of 5 OD at $\lambda_{max}$. After this round of

dilution, the final stock solution II has CTAB concentration of ~ 0.007 mM and the total volume of ~ 677 μL.

## Synthesis of symmetry-broken patchy prisms

Once stock solution II is prepared, 2-NAT solution (1 μL, 2 mg/mL in DMF) was mixed with 819 μL DMF in an 8 mL glass vial. 100 μL of stock solution II and 100 μL water were then sequentially added into the vial dropwise using pipette on the vortex, followed by addition of PS-*b*-PAA solution (80 μL, 8 mg/mL in DMF) in one shot, followed by gentle shaking on the vortex. The α of this reaction mixture, defined as the ratio of 2-NAT molar concentration to OD of the prisms at $\lambda_{max}$ is 25 nM. The vial was tightly capped with a Teflon-lined cap and sonicated for 5 s, parafilm-sealed, and heated throughout the reaction at 90 °C in an oil bath and left undisturbed for 2 h. The reaction mixture was then cooled down to room temperature in the oil bath, which typically took 80 min. The solution was transferred to a 1.5 mL microcentrifuge tube and centrifuged three times (2800 x g, 2220 x g, and 1700 x g for 15 min each) to separate the residual 2-NAT and PS-*b*-PAA from the patchy prisms. After each centrifugation, 1.48 mL of the supernatant was removed and the 20 μL sediment was re-dispersed with 1.48 mL water. After the third round of centrifugation, 10 μL of the sediment was diluted by 190 μL of water. The patchy prisms solution was then filtered through the centrifuge tube filters at 930 x g for 10 min to remove the micron-scale PS-*b*-PAA films, followed by removing 150 μL of supernatant. The 8 μL of sediment solution was drop-casted onto a TEM grid pretreated with oxygen plasma for 1 min at low radio frequency (RF) power and left dry in the air for 3 h. The remaining 42 μL sediment was diluted by adding 58 μL water into the microcentrifuge tube for further characterization and storage (stable for at least 2–3 months).

For tip-patch prisms with varied number of patches (Figs. 2d, f, 3a), reaction temperature $T$ was increased to 100 °C or 110 °C while keeping a constant α of 50 nM (Fig. 2d), or α was varied in range between 25 nM and 75 nM at constant $T$ of 90 °C (Fig. 2f), as detailed in Supplementary Table 1. The rest of the sample preparation and washing procedures were the same as the above for single-patch nanoprisms.

For the quenching reaction (Supplementary Fig. 5a), we kept the reaction condition the same as the experimental condition generating single-patch prisms in Fig. 1b (α = 25 nM and $T$ = 90 °C), while decreasing the reaction time by 90%. After 12 mins' of grafting reaction, we took out the reactor vial from the oil bath and quickly cooled it down to room temperature in the air, which took less than 5 min. The rest of the sample washing procedures were the same as the above for single-patch nanoprisms.

For the prisms coated by 2-NAT-only without polymers (Supplementary Fig. 5b), we kept the reaction condition the same as the experimental condition generating single-patch prisms in Fig. 1b (α = 25 nM and $T$ = 90 °C), except for replacing PS-*b*-PAA solution (80 μL, 8 mg/mL in DMF) to DMF (The final DMF to water ratio is 4.5 to 1). After the reaction, the solution was transferred to a 1.5 mL microcentrifuge tube and centrifuged three times (2800 x g, 2220 x g, and 1700 x g for 15 min each) to separate the residual 2-NAT from the prisms. After each centrifugation, 1.48 mL of the supernatant was removed and the 20 μL sediment was re-dispersed with 1.48 mL DMF. After the third round of centrifugation, 5 μL of the sediment was diluted by 20 μL of water. The 5 μL of sediment solution was drop-casted onto a silicon wafer pretreated with oxygen plasma for 1 min at low RF power and left dry in the air for 3 h.

The patchy prisms synthesized by multiple steps (Supplementary Fig. 5c) were prepared by sequentially adding 2-NAT and PS-*b*-PAA based on the experimental condition generating single-patch prisms in Fig. 1b (α = 25 nM and $T$ = 90 °C). First, 2-NAT solution (1 μL, 2 mg/mL in DMF) was mixed with 819 μL DMF in an 8 mL glass vial. 100 μL of stock solution II and 81.8 μL water were then sequentially added into the vial dropwise using pipette on the vortex. The vial was tightly capped with a Teflon-lined cap and sonicated for 5 s, parafilm-sealed, and heated throughout the reaction at 90 °C in an oil bath and left undisturbed for 2 h. After the reaction mixture was cooled down to room temperature in the oil bath, PS-*b*-PAA solution (80 μL, 8 mg/mL in DMF) and water (18.2 μL) were added into the vial, followed by gentle vortexing. The vial was tightly capped with a Teflon-lined cap and sonicated for 5 s, parafilm-sealed, and heated again at 90 °C in an oil bath for 2 h. The rest of the sample washing procedures were the same as the above for single-patch nanoprisms. For the reversibility test (Supplementary Fig. 5d, e), we kept the same reaction condition as for the single-patch prisms in Fig. 1b (α = 25 nM and $T$ = 90 °C). Once the reaction mixture in the vial is cooled down to room temperature, the same vial was brought into the 110 °C oil bath and left undisturbed for 2 h. The rest of the cooling, washing and sample preparation steps are the same as above for the one-step single-patch prism synthesis.

## Synthesis of patchy NPs for generalizability test

For the patchy prisms synthesized using different polymer chain lengths (Supplementary Fig. 13), we used the same reagent concentrations as for generating the single-patch prisms (α = 25 nM, 80 μL of PS-*b*-PAA solution, 8 mg/mL in DMF), while varying the polymer lengths from $PS_{154}$-*b*-$PAA_{51}$ to $PS_{230}$-*b*-$PAA_{49}$ and $PS_{394}$-*b*-$PAA_{58}$, at the varied grafting temperature $T$ = 90 °C and/or 110 °C. Note that the final DMF to water ratio in the reactor was slightly modified for each polymer (from 4.5:1 to 4.78:1 and 6.35:1, respectively) as guided by increasing PS to PAA block ratio, to mimic similar solvent quality used for the $PS_{154}$-*b*-$PAA_{51}$. The rest of the sample washing procedures were the same as the above for single-patch nanoprisms.

For the symmetry-broken patchy NPs synthesized using differently shaped NPs (Supplementary Fig. 12), we first synthesized the gold octahedron[17,62] and bipyramids[17,63] following a literature method via seeded-growth. The as-prepared bipyramids were purified to remove spherical impurities using the literature method[17,63]. The stock solution of octahedron and bipyramids were prepared to have 10 OD at $\lambda_{max}$ (578 nm and 730 nm, respectively) with 0.01 mM CTAB, by centrifuging (4990 x g, 2800 x g, 2220 x g for 16 min each) the as-synthesized octahedra and purified bipyramids in an aqueous CTAB solution (20 mM, containing NaI). The polymer coating reaction conditions were kept as α = 25 nM and $T$ = 90 °C and α = 1 nM and $T$ = 90 °C, for octahedra and bipyramids, respectively. The mixture in the vial was heated at 90 °C in an oil bath and left undisturbed for 2 h. The rest of the cooling, washing (three times of centrifugation at 4900 x g, 2800 x g, and 1700 x g for 15 min each) and sample preparation steps are the same as above.

## Single-patch nanoprism assembly

The as-prepared single-patch prism solution (~1.1 OD at $\lambda_{max}$, 300 μL) was mixed with CTAB (0.2 mM, 100 μL) in 1.5 mL microcentrifuge tube by gentle pipetting for 10 times. The mixture was kept undisturbed overnight. 300 μL of the supernatant was transferred to another 1.5 mL microcentrifuge tube and centrifuged at 1250 x g 10 min. After the centrifugation, 270 μL of the supernatant was removed and the 30 μL sediment was homogeneously mixed by gentle pipetting. The 8 μL of sediment solution was drop-casted onto a TEM grid pretreated with oxygen plasma for 1 min at low RF power and left dry in the air for 3 h.

The control assembly experiment using NaCl (Supplementary Fig. 20) was performed following a literature method[36]. First, on the clean side of a parafilm, single-patch prism solution (~1.1 OD at $\lambda_{max}$) was mixed with aqueous NaCl solution (200 mM, detailed amount listed in Supplementary Table 6) by gentle pipetting for 10 times. The mixture droplet was left undisturbed under a glass Petri dish for 5 min.

An oxygen plasma-treated TEM grid was placed up-side-down on the mixture and kept undisturbed for 30 min to allow single-patch prisms to assemble. To minimize the drying of the droplet, two vial caps filled with water were placed near the mixture under the Petri dish. After 30 min, the droplet beneath the grid was soaked out by gently touching the edge of the grid using a filter paper cut into sector-shape. The remaining liquid on the grid was further absorbed by using another piece of filter paper gently touching the grid, followed by grid dried in the air for 1 h.

The tri-patch prism assembly via surfactant-polymer complexation (Supplementary Fig. 21) was performed based on the two methods with a slight modification. The as-prepared tri-patch prism solution (1 OD at $\lambda_{max}$, 300 μL) was mixed with CTAB (0.2 mM, 100 μL) in 1.5 mL microcentrifuge tube by gentle pipetting for 10 times and left undisturbed for 15 min. The 10 μL of the mixture was drop-casted onto an oxygen plasma-treated TEM grid and left undisturbed under a glass Petridish. To minimize the drying of the droplet, two vial caps filled with water were placed near the mixture under the Petri dish. After 15 min, the droplet on the grid was soaked out by the filter paper. The remaining liquid on the grid was further absorbed by using another piece of filter paper gently touching the grid, followed by grid dried in the air for 1 h.

## Characterizations of patchy nanoprisms and their assembly

UV-Vis spectra of the prisms and separated impurities were measured using a Scinco S-4100 PDA spectrophotometer with a quartz cuvette (path length = 1 cm, VWR). Harrick Plasma PDC-32 (maximum RF power of 18 W, Harrick Plasma) was used for plasma treatment of TEM grids. Asylum Cypher S (Asylum Research) was used for AFM measurement of single-patch prisms. Confocal Raman spectroscopy was performed using a Nanophoton Raman 11 system for ligand-coated prisms in the liquid state at room temperature. A JEOL 2100 Cryo TEM with a LaB$_6$ emitter at 200 kV and Hitachi S-4800 SEM were used for characterizing the size and shape of the prisms, patchy prisms, and the assembled structures. Details regarding TEM image analysis are in Supplementary Notes 5 and 7.

## TEM Tomography of patchy nanoprisms and their assembly

A JEOL 2100 Cryo TEM at an acceleration voltage of 200 kV was used for TEM imaging of the patchy nanoparticles. Low electron dose rates (6–8 e$^-$Å$^{-2}$s$^{-1}$) were applied using spot size 3 to minimize beam-induced alteration. A tilt series of TEM images was collected over a tilt range of −60° to +60° with angle increment of 2° (total 61 images). Each image was collected with an exposure time of 1 s, resulting in a dose per image of 6–8 e$^-$Å$^{-2}$s$^{-1}$. Throughout all tilt series acquisition, the sample was set to its eucentric height at each tilt angle manually, followed by a defocus of −2048 nm to improve contrast. The collected TEM images were aligned using the patch tracking module in the open-source software IMOD[64] 4.9.3. Tomograms were generated using the Model-Based Iterative Reconstruction (MBIR) algorithm[65] with diffuseness of 0.3 and smoothness of 0.2. The segmentation of reconstructed 3D volumes of the patchy nanoparticles was performed in Amira[66] 6.7. A median filter over a 3 × 3 × 3 voxel neighborhood and 26 iterations was applied, followed by a 3D Gaussian filter with a kernel size of 9 in 3D with standard deviation voxel of 3 × 3 × 3 and then a 3D edge-preserving smoothing filter with the values 25 time, 5 step, 3.5 contrast, and 3 sigma. A grayscale threshold was set to generate an approximately segmented volume, which was then corrected using a manual adjustment.

## AFM measurements

AFM images of single-patch prisms (Supplementary Fig. 3c) were obtained using an Asylum Cypher S AFM (Asylum Research) in tapping mode. The AFM sample was prepared by dropcasting an aliquot (5 μL) of patchy prism suspension onto a small piece (~ 0.2 cm$^2$) of Si wafer, which was dried in the air overnight before the AFM measurements. Note that the Si wafer was pre-cleaned with acetone and isopropanol, and treated with oxygen plasma (1 min at low RF power) to make it hydrophilic.

## Raman characterization

The specimen for Raman characterization was freshly prepared before the measurement by dropcasting an aliquot (10 μL) of the 2-NAT coated prism suspension prepared above onto a glass slide. Note that the glass slide was pre-cleaned with acetone and isopropanol, and blow dried with N$_2$. The excitation wavelength for Raman measurement was 785 nm, and three scans (each with 300 s exposure time) were taken and averaged to get the final spectrum as shown in Supplementary Fig. 8. A 20× objective lens was used, and the laser power was adjusted to be within the range of 0.5–1 mW. All the measured spectra were calibrated using the emission lines of a neon lamp.

## Grafting theory and Monte Carlo Grafting Simulations

To account for chain–chain interaction, we introduce the classical Flory interaction parameter χ into our scaling argument for predicting chain partitioning on an anisotropic surface (see full derivation of our scaling theory in Supplementary Notes 1, 6, and 8). To carry out Monte Carlo grafting simulations (detailed in Supplementary Note 3), we start by initializing an occupancy matrix for all grafting sites on a triangular prism (generated via surface mesh of the prism surface). We then employ our derived scaling relationship to compute the probability of grafting a chain at each surface location. Selection of an attachment site for the first chain employs the traditional Metropolis algorithm. For each attached chain, we update the occupancy matrix at the corresponding surface location to prevent multiple chains from grafting to the same location. All surface sites within a correlation length ξ (derived in Supplementary Note 1) of occupied sites are assigned a non-zero χ to model chain–chain interactions. This process repeats until the targeted grafting density is reached.

## LSPR computation

We created computer models of a particular assembly configuration of the patch-merged nanoprisms using finite element modeling with a customized mesh shape and size (spatial resolution of 0.1 nm) to fit experimental observations in TEM images in Fig. 4a, Supplementary Figs. 17 and 18. These adaptive meshes implemented using COMSOL[67] 5.2 provide vastly superior control of the model architectures over grids with preset shapes, improving near-field accuracy and significantly decreasing computation time. The nanoprisms assembled into bowties and other various trimer and tetramer structures were modeled as prisms with side lengths taken from the corresponding particles using their TEM images, with the sides being truncated using COMSOL's fillet function. These assemblies were immersed in a domain to which we assigned the dielectric constant of air to mimic the TEM sample preparation environment. Perfectly matched layers (PML) are used to truncate the spatial domain (3D space) containing the nanostructures for solving the frequency domain form of Maxwell's equations using the finite element method. The PML domain which is minimally reflective is used to absorb all outgoing waves. A plane-polarized wave was incident normally upon these structures and the electric field components were calculated on the lateral directions. The absorption C$_{Abs}$ and scattering C$_{Sca}$ cross sections were computed for each particle in the bowtie and the trimer assemblies. The extinction cross-section C$_{Ext}$ was calculated as the sum of the aforementioned absorption and scattering at the cross-sections by the following equation: C$_{Ext}$ = C$_{Abs}$ + C$_{Sca}$. The efficiency factors were determined by dividing the optical cross-sections with the physical cross-section of the specific nanoprisms. The simulations were terminated when the number of iterations reached a linear error value plateauing to zero

($-10^{-8}$). Material properties were modeled using the refractive indices of gold in the spectral range of 400–600 nm from the Johnson and Christy online database[68]. The relative electric field distribution maps are color-coded by the intensity enhancement for all the structures shown in Fig. 4g, Supplementary Figs. 17 and 18. The interpretations of LSPR computation for structures are further discussed in Supplementary Note 9.

## Reporting summary

Further information on research design is available in the Nature Research Reporting Summary linked to this article.

## Data availability

Experimental data (EM images and statistics) are provided in the figures and Supplementary Information. Simulation data associated with this study can be found at https://github.com/VoGroupJHU/patchy_grafting. Additional data are available from the corresponding authors upon request.

## Code availability

Custom MATLAB codes for MC simulation and analysis of simulation results are provided at https://github.com/VoGroupJHU/patchy_grafting. Additional codes can be available upon request.

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

## Acknowledgements

Synthesis and self-assembly experiments for this work was supported by the U.S. Department of Energy, Office of Basic Energy Sciences, Division of Materials Sciences and Engineering, under Award DE-SC0020723 (A.K. and Q.C.). Experiments were carried out in part in the Materials Research Laboratory Central Research Facilities, University of Illinois. Theory and simulation for this work was supported by the Department of the Navy, Office of Naval Research under ONR award number N00014-18-1-2497 (T.V. and S.C.G). This research utilized computational resources and services supported by Advanced Research Computing at the University of Michigan, Ann Arbor, and provided by the Extreme Science and Engineering Discovery Environment (XSEDE), which is supported by National Science Foundation Grant ACI-1053575, XSEDE Award DMR 140129 (T.V. and S.C.G.). LSPR of this work was performed by P.B. at the Center for Nanoscale Materials, a U.S. Department of Energy Office of Science User Facility, and was supported by the U.S. DOE, Office of Basic Energy Sciences, under Contract No. DE-AC02-06CH11357. D.M. and P.B. acknowledge partial support from the Center for Dynamics and Control of Materials: an NSF Materials Research Science and Engineering Center (NSF MRSEC) under Cooperative Agreement DMR-1720595 and the Welch Foundation (F-1848). A.K and T.V. thank Prof. Kenneth S. Schweizer and Prof. Catherine J. Murphy for helpful discussions.

## Author contributions

A.K. and Q.C. designed the experiments. A.K., L.Y., C.K., and Q.C. performed the experiments and data analysis. T.V. developed the theory, performed MC simulations, and analyzed the simulation data. H.A. performed electron tomography and S.Z. performed AFM and Raman measurement. P.B. and D.M. performed FEM calculation. All authors contributed to the writing of the paper. Q.C. and S.C.G. supervised the work.

## Competing interests

The authors declare no competing interests.
