## [Peer Review File · Nature Communications]

Symmetry-breaking in patch formation on triangular gold nanoparticles by asymmetric polymer graftingREVIEWER COMMENTS

Reviewer #1 (Remarks to the Author):

The manuscript "Symmetry-breaking in patch formation on gold nanoparticles via supramolecular "bandwagoning" by Chen, Glotzer and coworkers describe the observation of preferential adsorption of polymer chains to tips of gold nanoprisms. The authors show convincing evidence through a large number of experiments and theoretical models that there indeed seems to be a selective decoration of a single tip out of the three depending on the preceding chemical modification of the tips with an aromatic thiol (2-NAT). The authors find that by addition of PS-b-PAA block copolymers, chains do accumulate on one tip first explained by a chain-chain attraction through vdW interaction and support this find by a theoretical model as well as simulations. The authors provide comprehensive data sets thereby scrutinizing several control parameters and their effect on patch, size, shape, distribution and so on. Finally, aggregates of prisms are discussed when increasing the nanoprism concentration.

While I do find the effect convincing and interesting, there are a couple of issues and comments:

Main issues:

1. Explanation of the underlying effect

The authors find that by addition of PS-b-PAA block copolymers, chains do accumulate on one tip first, which is explained by a chain-chain attraction through vdW interaction. This explanation is rather surprising, because I do not see how the first polymer-decorated tip should influence accidental collision of other chains with the second tip in 60 nm distance. I would assume that a coiled chain could collide with the second tip at any time and start attracting further chains there as well. Can the authors give an explanation why the second tip should not be decorated as drawn in the schematic in Fig. 2a? What is the expected hydrodynamic radius of a coiled chain and can it be deflected from tip 2 towards tip 1 over a distance of 65 nm to suppress chain-tip collision through random diffusion?

The authors further explain that the local concentration of the block copolymer at the tips needs to be controlled, which makes sense, but then the authors do this by changing the amount of added 2-NAT per nanoprism (α). However, less 2-NAT also changes the hydrophobicity of the tip and the size of the grafting site as the thiol should attach to the most reactive planes of the gold particle (probably closest to the tip?). Maybe it is the amount of 2-NAT that controls the number of reactive sites per prism, i.e. attached 2-NAT preferentially attracts more 2-NAT to the same location thereby creating single-tip modification? Can the authors rule this theory out over the theory of polymer chains attracting other polymer chains?

The authors explain that "Varying α controls the chain concentration locally available at the prism tip and is related to the grafting density". While I agree that the size of the 2-NAT patch may control available space for chain adsorption, the term grafting density implies knowledge of how chains attach. Unlike for covalently attached chains (like the authors draw in Fig. 2a), several PS units could also lie flat on the surface and interact with several naphthalene units making a grafting density more or less a wild guess. Could the authors give experimental evidence about their theory of chain attachment to the surface?

2. Disappointing ending

The authors do form aggregates with their well-controlled nanoprism. Unfortunately, this is done simply by increasing the particle concentration to enhance the probability for patch-patch collision but with the consequence of rather random structures (and low yield). The manuscript would become much more impactful, if the previously hard-earned control of patch distribution was actually used to form defined aggregates in high-yield. Why did the authors not trigger aggregation on purpose, e.g. by addressing the PAA surface-decoration of the tips (PAA should face towards the solution) and add multivalent cations (Ca^{2+} , Fe^{3+} , polycations polymers or the like)? I really suggest the authors consider this approach or another one to greatly improve their formed aggregates to raise this final part of the manuscript to the high level of the rest and the appropriate level for Nature Communication.

Other issues:

1. Some particle statistics are done on odd numbers, e.g. caption of fig. 1 states that 272 and 97 particles are counted. Why not 300 and 100, or better yet, 500 particles for each species as is good practice in colloid science.

2. The authors state “At low grafting temperature T , chain–chain attraction drives strong localization, producing symmetry breaking patches (Fig. 2b).” However, temperature does not seem to be a very effective control parameter, as prisms have mostly 2-3 patches, whereas 3 are kind of the expected given the energetically beneficial tips (enthalpic and entropic) vs. the rest of the surface.

3. I do not find the analogy to “bandwagoning” appropriate (although I was not aware of this expression before). “The approach is analogous to the bandwagon effect—a phenomenon whereby individuals are swayed through human interactions to adopt the same traits as their influencers”. The abstract is even worse “Asymmetric grafting leads to selective surface partitioning of polymers in a manner resembling the bandwagon effect, whereby individuals adopt particular behaviors, mannerisms or views simply because others have adopted them.” Polymers and particles aggregate simply by reduction of free energy and not a psychologic or intelligent decision. This “lurid headlines” should be reconsidered for more scientific, down-to-earth expressions.

Overall, the authors show great expertise in the field and have compiled a large data set to support their results. Although I am not fully convinced by the provided explanation (and certainly not a fan of alleged ascribed intelligence) , the data is there and was carefully analyzed. Since I do not see any technical issues to address and the observed effect is certainly interesting, I recommend this manuscript for publication as Article in Nature Communication after the authors have addressed the raised questions and comments.

Reviewer #2 (Remarks to the Author):

In this manuscript by Chen et al. the authors demonstrated the formation of symmetry-breaking polymer patches on gold nanoprisms through the so-called supramolecular bandwagoning effect. The gold nanoprisms were first modified with a hydrophobic small molecular thiol ligand (2-NAT) then PS-b-PAA can be adsorbed into the tip of nanoprisms due to the hydrophobic interaction. By tuning the temperature of thermal annealing and the concentration of 2-NAT, the single and double patchy nanoprisms can be generated at a high yield. The experiments agreed well with the proposed theory, which disclosed the mechanism of patch size and shape control. The asymmetric patchy nanoprisms can undergo patch merge and give colloidal bowties. The manuscript is well-written and the results are very interesting in the field of patchy nanoparticles, thus I would recommend its publication. However, the following questions must be addressed:

1. For the single- and double-patchy nanoprisms, although the PS-b-PAA was centralized into one or two specific tips due to the supramolecular bandwagoning effect, was the uncovered tip still chemically grafted with hydrophobic 2-NAT ligand? If so, how were they stabilized with two or one hydrophobic tips? Why didn't they assemble through a tip-tip contact manner?

2. For the colloidal bowties, in what conditions were they formed? What is the configuration of PAA after the merging of PS soft patches? Is it possible to assemble them into nanoprism-networks through patch merging of three-tip patched building blocks?

Reviewer #3 (Remarks to the Author):

In “Symmetry-breaking in patch formation on gold nanoparticles via supramolecular “bandwagoning””, the authors an elegant procedure for patchy polymer-functionalized gold nanoparticles. The authors carefully analyzed the functionalization of gold nanotriangles by a combination of naphthalenethiol and

PS-PAA. I believe that this manuscript is of interest to the scientific community and could be published after the authors have addressed a few points.

The mechanism proposed by the authors supposed that the chemisorption of the NAT ligand occurs on all tips. Could this be quantified using EELS? Alternatively, can the gold nanoparticle be first functionalized with NAT and then PS-PAA after purification? The one-pot strategy, while practical, raise the question?

The authors claim that the strategy proposed is general. However, only one particle/polymer couple is studied. To strengthen their claim of a generalized strategy, the authors should investigate the formation of patches on other particles (nanorods?) and with at least one other copolymer.

Can the authors comment on the reversibility of the patches created? That is, if a particle with 1 patch prepared at low temperature, is purified and then incubated at a higher temperature will the morphology be preserved? Or if the solvent quality is changed?

Can the authors describe in more detail the formation of the bowtie assemblies and how the assembly process can be controlled. For example, does the concentration of NPs in suspension affect the fraction of NPs undergoing assembly?

Can the authors provide the number of particles analyzed in tables 2-3 in supporting information.

Responses to Reviewer 1:

Reviewer 1 commented that we “*show great expertise in the field and have compiled a large data set to support their results*”, “*the observed effect is certainly interesting*”, and thus “*recommend this manuscript for publication as Article in Nature Communication after the authors have addressed the raised questions and comments.*” We thank the reviewer for the questions and comments to make our manuscript stronger, all of which we have addressed using new experiments and analysis (revised main text Fig. 1, new main text Fig. 5, Supplementary Notes 2, 4, Supplementary Figs. 3, 5, 9, 13, 19, 20), and new discussions as detailed below, particularly on clarifying the symmetry-breaking patch formation mechanism and introducing a new method to significantly increase the yield of tip-to-tip assembly structures.

Comment 1: “*The authors find that by addition of PS-*b*-PAA block copolymers, chains do accumulate on one tip first, which is explained by a chain-chain attraction through vdW interaction. This explanation is rather surprising, because I do not see how the first polymer-decorated tip should influence accidental collision of other chains with the second tip in 60 nm distance. I would assume that a coiled chain could collide with the second tip at any time and start attracting further chains there as well. Can the authors give an explanation why the second tip should not be decorated as drawn in the schematic in Fig. 2a? What is the expected hydrodynamic radius of a coiled chain and can it be deflected from tip 2 towards tip 1 over a distance of 65 nm to suppress chain-tip collision through random diffusion?*”

Response: We thank the reviewer for helping us clarify the underlying asymmetric grafting effect. We agree that the distance between two neighboring tips (62.5 nm) is greater than the coiled chain size of 9.3 nm (assuming good solvent). However, two things are worth noting. Firstly, the size of a chain decorating the tip of the prism (22.9 nm) is generally more extended than that of a free coil because an extended configuration is entropically more favorable, as demonstrated in prior studies (*ACS Nano* **13**, 14241 (2019); *Sci. Adv.* **5**, eaaw2399 (2019)), given the additional room around a tip vs. an edge or face. The increased extension implies that the interaction range between a free chain in solution and the chain decorating the tip can effectively increase. Secondly, chains can dynamically reorganize. Even if there are chains stochastically adsorbed on a second tip, they can detach into the solution or move around the particle surface to find the most energetically favorable sites. In other words, as long as the maximal extension length of one chain falls within the maximal range of the other, the chains can sample many possible configurations and reorganize to an *equilibrium* single patch formation, as confirmed via experimental observations of high-yield single-patch prisms (Fig. 1) and simulations of the polymer adsorption process (Fig. 2h–i).

In the revised manuscript, we added a new Supplementary Fig. 9 showing the number density of Kuhn monomers of two chains when they adsorb on neighboring tips, added discussions and explanations of Supplementary Fig. 9 in Supplementary Note 4, and added discussions in the main text to elaborate the discussion on the chain–chain attraction-induced asymmetric grafting mechanism. Specifically:

- (1) We added a new Supplementary Fig. 9. In this figure, we plotted the Kuhn monomer density distribution, $\phi(r)$ (predicted from theory), as a function of its distance from the tip. One chain is placed at a reference tip ($r = 0$) and the other at a distance equal to the side length of the prism (R_{tip}). The gray shading region corresponds to the space where monomers on the two chains overlap. Even though the two chains are placed at the two tips of a 62.5 nm distance away, the monomers within this region can experience strong polymer–polymer interaction that drive chain relocation and subsequent merging to a single tip.

Supplementary Fig. 9. Monomer density distribution $\phi(r)$ prediction as a function of distance from the tip. As detailed in Supplementary Note 4, one chain is placed at a reference tip ($r = 0$) and the other at a distance equal to the side length of the prism (R_{tip}). Single-dashed lines mark the equilibrium, hydrodynamic size of the tip-decorating chain (R_{chain}) and double-dashed lines indicate the fully extended size of the chain (R_{rigid}). Colors correspond to the same chain (blue: chain at reference tip, orange: chain at adjacent tip) and the x -axis is scaled by R_{chain} . The gray shading region indicates the space where the chains' monomer densities overlap with each other. This means that any monomer within this region will experience strong monomer-monomer interactions that ultimately drive chain reorganization and subsequent merging to a single tip.

(2) We added the following discussion to the main text:

“We note that a second prism tip can become decorated with polymer chains during the grafting process. But, because grafting is a stochastic process, those chains can subsequently detach back into solution and continue to sample the prism surface to find the most favorable site—the tip already occupied by other chains—to maximize the polymer–polymer attraction of the whole system, as also guided by their long interaction range on the tips (Supplementary Note 4 and Supplementary Fig. 9). This equilibration process is fast, as modeled by an obvious preferential adsorption kinetics in our simulation (Fig. 2h–i).”

Comment 2: “The authors further explain that the local concentration of the block copolymer at the tips needs to be controlled, which makes sense, but then the authors do this by changing the amount of added 2-NAT per nanoprism (α). However, less 2-NAT also changes the hydrophobicity of the tip and the size of the grafting site as the thiol should attach to the most reactive planes of the gold particle (probably closest to the tip?). Maybe it is the amount of 2-NAT that controls the number of reactive sites per prism, i.e. attached 2-NAT preferentially attracts more 2-NAT to the same location thereby creating single-tip modification? Can the authors rule this theory out over the theory of polymer chains attracting other polymer chains?”

Response: We thank the reviewer for this point! In the original manuscript, we considered the possibility that the amount or concentration of 2-NAT controls the number of tips coated by 2-NAT and thus the number of polymer-coated tips. However, we ruled this possibility out based on our control experiments:

(i) the adsorption behavior of 2-NAT on the gold NP surface does not change over temperature variations of 90–110 °C as suggested by our Raman characterization, “Raman spectra of the prisms show a constant peak intensity for the 2-NAT ligand³⁹ over the range of temperature (90–110 °C) (Methods, Supplementary Fig. 8a), suggesting negligible change in the 2-NAT ligand adsorption on NPs⁴⁰.” (ii) Over the entire 2-NAT concentration range we used in this work including that leads to single-patch prism synthesis (Fig. 1), at a grafting temperature of 110 °C, we consistently obtained tri-patch prisms, suggesting that this 2-NAT concentration range allows its adsorption onto all the three prism tips (Supplementary Fig. 2): “Notably, increasing to 110 °C induces symmetric grafting with a 99% yield of triple-patch NPs, consistent with our previous report³⁵.” “Therefore, the transition to triple-patch grafting at 110 °C is a result of weakened chain–chain attraction that produces equipartitioning of chains across all prism tips.” Thus we conclude that lower temperature induces stronger polymer–polymer attraction which drives the formation of single- or double-patches. This observation is also *quantitatively* supported by the polymer scaling theory and simulations. This conclusion is consistent with the fact that 2-NAT has smaller molecular weight (160 g/mol) and thus smaller intermolecular attraction in good solvent condition than the extended polymer chains (20,000 g/mol).

We agree with the reviewer that this is an important point to highlight and illustrate. In the revised manuscript, we further clarified this discussion and added a series of new experiments (Supplementary Figs. 5, 13) as follows:

- (1) We performed new experiments to show that the prisms are coated with 2-NAT at all tips at 90 °C. First, we synthesized prisms that are processed at the same condition as that used for the single-patch prism presented in Fig. 1, but without the polymers. We find that these prisms, when dried, can assemble tip-to-tip connected through all the three tips (Supplementary Fig. 5b), suggesting a balance of tip-to-tip hydrophobic attraction (due to 2-NAT coating) and electrostatic repulsion due to the residue charged ligands on the prism facets. Second, when we then performed the subsequent experiment to incubate these 2-NAT coated prisms with polymers, we still obtained the single-patched prisms (Supplementary Fig. 5c), corroborating again that the asymmetric patch formation originates from the effect of polymer–polymer attractions.

Supplementary Fig. 5b,c. 2-NAT decorates all three tips. (b) SEM image of non-patch nanoprisms decorated with 2-NAT only, without polymers ($\alpha = 25$ nM, same as that in the single-patch prism synthesis condition in Fig. 1, see Methods), assembling by their tips due to hydrophobic attraction between 2-NAT covered tips. (c) Sequential addition of PS-*b*-PAA leading to asymmetric patch grafting despite 2-NAT covering all three tips. Scale bars: 200 nm (b), 100 nm (c).

- (2) We also performed new experiments that, at identical grafting conditions (e.g., DMF/water ratio, 2-NAT concentration, grafting temperature), higher molecular weight polymers (longer PS block) than those used in the main text (PS₁₅₄-*b*-PAA₅₁) lead to higher polymer–polymer attraction, and indeed promote asymmetric patch decoration. Specifically, as shown in Supplementary Fig. 13b,

at 110 °C, the relatively short polymer (PS₂₃₀-*b*-PAA₄₉) led to tri-patch prisms. In comparison, when we keep all conditions the same but just use a longer polymer (PS₃₉₄-*b*-PAA₄₉), single-patch prisms are formed (Supplementary Fig. 13d). This observation is consistent with the molecular weight effect predicted by the scaling theory (Fig. 2j, Supplementary Fig. 13c). In these experiments, the 2-NAT concentrations are kept the same, again showing that it is not the preferential adsorption of 2-NAT that leads to asymmetric patch decoration.

Supplementary Fig. 13. Extension of the “bandwagon effect” to polymers of different chain lengths. (a,b) TEM images of single- and tri-patch nanoprisms obtained using PS₂₃₀-*b*-PAA₄₉ at $T = 90\text{ °C}$ (a) and $T = 110\text{ °C}$ (b) at a fixed 2-NAT concentration of $\alpha = 25\text{ nM}$. (c) Phase diagram of patch count n (shading colored to n as noted in the legend) across the parameter space: grafting temperature T , chain length N , and ligand concentration α . The phase diagram is partitioned to different colored regions following simulation results. The black lines are phase boundaries predicted by theory. Arrows show illustrations of estimated corresponding experimental variation in (a,b,d). (d) TEM images of single-patch nanoprisms obtained using PS₃₉₄-*b*-PAA₅₈ at $T = 110\text{ °C}$ and $\alpha = 25\text{ nM}$. Scale bars: 100 nm.

(3) We added the corresponding discussions in the main text based on these new experimental results.

“Notably, increasing to 110 °C induces symmetric grafting with a 99% yield of triple-patch NPs, consistent with our previous report³⁵. Raman spectra of the prisms show a constant peak intensity for the 2-NAT ligand³⁹ over the range of temperature (90–110 °C) (Methods, Supplementary Fig. 8a), suggesting negligible change in the 2-NAT ligand adsorption on NPs⁴⁰. Therefore, the transition to triple-patch grafting at 110 °C is a result of weakened chain–chain attraction that produces equipartitioning of chains across all prism tips. This observation extends to the range of

α from 25 nM to 75 nM that we use in this work where, at 110 °C, tri-patch prisms are consistently obtained (Supplementary Fig. 2a–d). These results emphasize systematic tunability of the supramolecular bandwagon effect induced by strong chain–chain interaction to control the number of patches and thus the symmetry of the patchy prisms, despite the fact that all three tips of prisms are coated by 2-NAT. This is also consistent with our polymer scaling theory prediction that—as compared to the polymer chains of molecular weight 20,000 g/mol—short ligands of 160 g/mol (i.e., 2-NAT) are not necessarily preferentially accumulated on one tip.”

“In a sequential control experiment where the prisms are first incubated with 2-NAT, harvested, and then incubated with polymers, the prisms out of incubation with 2-NAT only can assemble into structures due to the hydrophobic attractions at all three tips (Supplementary Fig. 5b). Nevertheless, despite the 2-NAT decoration on all the three tips, the subsequent polymer grafting upon polymer addition generates single-patch prisms (Supplementary Fig. 5c), confirming that strong chain–chain interaction induces asymmetric grafting.”

“On the other hand, using PS₂₃₀-*b*-PAA₄₉, we see similar results as for PS₁₅₄-*b*-PAA₅₁. At a low grafting temperature of 90 °C, single-patch prisms are obtained whereas increasing temperature to 110 °C produces tri-patch prisms (Supplementary Fig. 13d). Interestingly, by using even longer polymer chains PS₂₃₀-*b*-PAA₄₉, we can get single-patch prisms even at 110 °C. This transition is also consistent with the phase diagram (Fig. 2i) as one move towards the right along the *x*-axis.”

Comment 3: “The authors explain that “Varying α controls the chain concentration locally available at the prism tip and is related to the grafting density”. While I agree that the size of the 2-NAT patch may control available space for chain adsorption, the term grafting density implies knowledge of how chains attach. Unlike for covalently attached chains (like the authors draw in Fig. 2a), several PS units could also lie flat on the surface and interact with several naphthalene units making a grafting density more or less a wild guess. Could the authors give experimental evidence about their theory of chain attachment to the surface?”

Response: We thank reviewer for pointing out this important detail. The key experimental evidence for the polymer chains adopting an end-attached configuration on the NP surface is the patch thickness (t_m), which is as much as 18.9 ± 4.0 nm (see Fig. 3a,d) across the conditions we have sampled in our work. This thickness matches with the expected extended chain length based on the polymers’ molecular weight (~22.9 nm, in good solvent). Furthermore, we note that the solvent quality and PAA block fraction ϕ_A used is within the micellar formation regime (i.e., low ϕ_A regime on the block co-polymer phase diagram; *Chem. Soc. Rev.* **41**, 5969 (2012); *Small* **7**, 2721 (2011)). On surfaces, this block copolymer will exhibit an equilibrium conformation of a double layer—PS inner layer and PAA outer layer. As discussed on our reply to Comment 1, the chains are mobile on the particle surface and thus we assume that they can reorganize into the “end attachment” double layer motif shown in the schematic of Fig. 2a. Since our theory is equilibrium by design, this assumption of the final “end attachment” conformation becomes appropriate for our geometrical scaling balances.

We also agree with the reviewer that when the polymer concentration on the prism surface (i.e., grafting density) is low, several PS units could lie flat on the NP surface, as shown in the NPs collected at 10% reaction procession (Supplementary Fig. 5a). In those cases, the patch thickness is smaller ($t_m = 5.8 \pm 2.3$ nm) due to the more flattened polymer configuration. However, when the system is in equilibrium, PS-*b*-PAA polymers form a double layer with their PS end attached on the prism surface. This configuration in polymer scaling theory still holds the polymer chains to be effectively attached by one end (PS), although the “end” becomes larger as the number of attached PS units is increased.

In the revised manuscript, we address this comment by adding the above detailed discussions on why the polymers are mostly end-attached in our experiments and why the end-attached configuration is sufficient for our theoretical modeling in Supplementary Note 2, and added the following discussions in the main text.

“The end-to-end distance R of an end-attached polymer chain (i.e. patch size, Supplementary Note 2) as a function of surface position Ω is derived to be: $R \sim r_0 \alpha^{1/5} v_0^{1/5} [1 - 2\chi]^{1/5} b_l^{2/5} \left[\frac{N b_l}{\Omega r_0} \right]^{3/5}$, where r_0 is the NP in-sphere radius, v_0 is the excluded volume of a Kuhn monomer of size b_l , and N is the number of Kuhn monomers in a chain.”

Comment 4: “*The authors do form aggregates with their well-controlled nanoprism. Unfortunately, this is done simply by increasing the particle concentration to enhance the probability for patch-patch collision but with the consequence of rather random structures (and low yield). The manuscript would become much more impactful, if the previously hard-earned control of patch distribution was actually used to form defined aggregates in high-yield. Why did the authors not trigger aggregation on purpose, e.g. by addressing the PAA surface-decoration of the tips (PAA should faces towards the solution) and add multivalent cations (Ca²⁺, Fe³⁺, polycations polymers or the like)? I really suggest the authors consider this approach or another one to greatly improve their formed aggregates to raise this final part of the manuscript to the high level of the rest and the appropriate level for Nature Communication.*”

Response: We appreciate the reviewer’s suggestion to make our work more impactful. In the original manuscript, the occasionally assembled bowties indeed have low yield.

In the revised manuscript, we added substantial new experimental results and presented a new assembly strategy based on surfactant-induced polyelectrolyte complexation (*Curr. Opin. Colloid Interface Sci.* **32**, 11 (2017); *Polymers* **11**, 51 (2019)). This is a method employed in making smooth and coherent polymer films (*Appl. Surf. Sci.* **458**, 903 (2018)) and in nucleotide purification (*Nat. Protoc.* **1**, 2320 (2006)) but has not been used in patchy particle assembly. It utilizes a positively charged surfactant molecule such as CTAB, which can have their charged heads adsorbed to and accumulated on the negatively charged PAA block of the polymer patch. The accumulated CTAB molecules lead to aggregation of the hydrophobic CTAB chains, which bridges the patches of nearby NPs to drive directed assembly. This complexation effect is exclusive to polymer patches and does not cause prism aggregation via their sides and faces.

We added a new main text Fig. 5, a new paragraph of discussion in the main text, new details in the “Method” section, and two new Supplementary Figures (Supplementary Figs. 19 and 20) to elaborate the details of this strategy, which significantly increased the yield of selective, patch-to-patch-only assembly to 80%. Furthermore, we verify the critical role of surfactants in this selective patch-patch complexation, compared to the previously reported assembly strategy using NaCl to simply screen the charge and thus not lead to patch-to-patch assembly.

(1) We added the description of the new assembly procedure and a new Fig. 5 in the main text, as well as a new Supplementary Fig. 19 for more TEM images.

“To further increase the yield of selective, patch-to-patch assembly that enhances plasmonic coupling, we develop a new assembly method based on surfactant-induced complexation, which has been employed for polymer film formation⁵⁴ and nucleotide purification⁵⁵ but not for directing NP assembly. According to previous studies^{56,57}, cationic surfactants (e.g., CTAB) facilitate aggregation of anionic polyelectrolytes (e.g., PAA) by forming a surfactant-polyelectrolyte complex (Fig. 5a,b). The head group of CTAB can attach onto the polyelectrolytes by electrostatic attraction, followed by hydrophobic attraction among the tails driving the aggregation of polymers

to which those tails are attached. Using this method for our single-patch prisms, we obtain 68% dimeric assemblies (376 structures counted, Fig. 5c–e, Supplementary Fig. 19) and 21% unassembled individual prisms. Importantly, among the dimeric assemblies, over 80% are bowties attached by the tip patches. Thus, this result shows greatly increased yield compared to the random patch merging shown in Fig. 4., implying patch-patch selective assembly.”

Figure 5. Patch-to-patch selective assembly of single-patch prisms. (a) Schematics of the CTAB-induced patch complexation mechanism. (b) Chemical structure of PS-*b*-PAA and CTAB involved in patch-patch selective assembly. (c) Low magnification TEM and SEM (inset) images of dimers. Yellow arrows highlight the merged patch interface. (d) High magnification TEM images of patch-patch assembled dimers. (e) Yields of assembly motifs in dimers. The notations in the *x*-axis mean as follows: n-n: between non-patched prism tips. p-s: patch to prism side. s-s: between prism sides. p-t: patch and non-patched tip. p-p: between patches. A total of 376 structures are counted. Inset: the yield of dimers compared to assemblies of trimers or larger (noted as ‘≥ 3’) and non-assembled NPs (noted as ‘1’). Scale bars: 100 nm (c), 50 nm (d).

Supplementary Fig. 19. TEM images of CTAB-induced patch complexation of single-patched prisms into patch-patch selectively assembled dimers. Scale bars: 100 nm (a), 500 nm (c–e).

- (2) We added the comparison experiment where we only add NaCl to our single-patch prisms to direct the assembly (the new Supplementary Fig. 20) and discussed the result in the main text.

“Additionally, when we use NaCl solution (concentration varied from 22.2 mM to 85.7 mM) and simply screen the charge-charge repulsion among patches, we do not obtain patch-to-patch assembly. Instead, the single-patch prisms are assembled through attachment of their non-coated gold surfaces (via sides or facets) as governed by strong van der Waals attraction between the gold (Supplementary Fig. 20), further highlighting the critical role of surfactant in favoring selective patch-patch assembly.”

Supplementary Fig. 20. NaCl induced assembly of single-patched prisms. (a) Schematics of the dimeric assembly motifs, governed by van der Waals force between gold prism surfaces. (b,c) Low magnification TEM images (b) and high magnification TEM images (c) of NaCl induced assembly at 22.2mM. (d,e) TEM images of NaCl induced assembly at 40 mM (d) and 85.7 mM (e). Scale bars: 50 nm.

Comment 5: “Some particle statistics are done on odd numbers, e.g. caption of fig. 1 states that 272 and 97 particles are counted. Why not 300 and 100, or better yet, 500 particles for each species as is good practice in colloid science.”

Response: In the revised manuscript, we increased the number of NPs we analyzed and the single-patch prism statistics in Figs. 1 and 2, which show no qualitative difference from the results presented in the original manuscript but increase confidence in our results. We updated the descriptions accordingly in the main text, captions of Figs. 1d,e, 2g, and Supplementary Table 3. We included a few more TEM images used in analysis as a new Supplementary Fig. 3d.

Supplementary Fig. 3. Single-patch prism dimensions. (d) Low magnification TEM images of single-patch prisms. Scale bars: 200 nm.

Comment 6: *“The authors state “At low grafting temperature T , chain–chain attraction drives strong localization, producing symmetry breaking patches (Fig. 2b).” However, temperature does not seem to be a very effective control parameter, as prisms have mostly 2-3 patches, whereas 3 are kind of the expected given the energetically beneficial tips (enthalpic and entropic) vs. the rest of the surface.”*

Response: We thank the reviewer for helping us clarify this point. We note that the temperature series presented in Fig. 2b are not at a ligand concentration optimized to obtain single-patch prisms at 90 °C. As a result, we rely on the statistical analysis presented in Fig. 2e to show that, as temperature increases, the fraction of evenly coated tri-patch particles increases. An obvious role of the temperature effect was instead shown in Fig. 1 and Supplementary Fig. 2. In Fig. 1, at 90 °C, single-patch prisms were obtained, while at 110 °C, tri-patch prisms were obtained while keeping all other conditions the same (Supplementary Fig. 2a).

In the revised manuscript, we emphasize further this strong temperature-only effect as follows:

“As shown in Fig. 2d and Supplementary Fig. 2a–d, tri-patch prisms are consistently obtained at 110 °C regardless of α , while asymmetrically grafted single-patch (Fig. 1) and two-patch prisms (Fig. 2d) are formed at 90 °C. Similarly, in our phase diagram (Fig. 2j), increasing temperature along the y-axis alone can control the patch number across a range of α , emphasizing the grafting temperature as an effective knob by controlling chain–chain attraction, and thus the patch grafting symmetry.”

Comment 7: *“I do not find the analogy to “bandwagoning” appropriate (although I was not aware of this expression before). “The approach is analogous to the bandwagon effect—a phenomenon whereby individuals are swayed through human interactions to adopt the same traits as their influencers”. The abstract is even worse “Asymmetric grafting leads to selective surface partitioning of polymers in a manner resembling the bandwagon effect, whereby individuals adopt particular behaviors, mannerisms or views simply because others have adopted them.” Polymers and particles aggregate simply by reduction of free energy and not a psychologic or intelligent decision. This “lurid headlines” should be reconsidered for more scientific, down-to-earth expressions.”*

Response: We appreciate the sincere comment and suggestion. In this manuscript, we used this terminology as a reader-friendly analogy to emphasize strong chain–chain attraction induced asymmetric grafting, as a new symmetry-breaking mechanism. We hope to keep the analogy at this level.

Responses to Reviewer 2

Reviewer 2 commented that “*The manuscript is well-written and the results are very interesting in the field of patchy nanoparticles, thus I would recommend its publication.*” We thank the reviewer for the positive feedback and also for the constructive comments to make our manuscript more clear and stronger, all of which we have addressed using new experiments and more analysis (new main text Fig. 5 and four new Supplementary Figs. 5, 19–21), new discussions and references detailed below.

Comment 1: “*For the single- and double-patchy nanoprisms, although the PS-*b*-PAA was centralized into one or two specific tips due to the supramolecular bandwagoning effect, was the uncovered tip still chemically grafted with hydrophobic 2-NAT ligand? If so, how were they stabilized with two or one hydrophobic tips? Why didn't they assemble through a tip-tip contact manner?*”

Response: Indeed as the reviewer commented, over the entire 2-NAT concentration range we used in this work (α : 25 to 75 nM), all three prism tips are covered by the hydrophobic 2-NAT ligands. We discussed this point in the original manuscript and added new control experiments to further corroborate this point (see our detailed response to Comment 2 of Reviewer 1). As to the colloidal stability of single- or double-patch prisms, they have negatively charged patches as the PAA blocks are pointing outwards, which can stabilize the patchy particles through electrostatic repulsion without forming into aggregation (for at least 2 months).

In the revised manuscript, inspired by the reviewer's comment, we performed a new experiment using the prisms coated only with 2-NAT (at 90 °C without polymers, at a condition that would otherwise lead to the formation of single-patch prisms in Fig. 1). As shown in the new Supplementary Fig. 5b, in the absence of charged polymer patches, the prisms are indeed observed to aggregate through their tips due to the coated 2-NAT providing hydrophobic attraction at all three tips.

In the main text, we added a new Supplementary Fig. 5b on this result, the discussions below, and the experimental details in the “Methods” section:

“In a sequential control experiment where the prisms are first incubated with 2-NAT, harvested, and then incubated with polymers, the prisms out of incubation with 2-NAT only can assemble into structures due to the hydrophobic attractions at all three tips (Supplementary Fig. 5b).”

Supplementary Fig. 5b. 2-NAT decorates all three tips. SEM image of non-patch nanoprisms decorated with 2-NAT only, without polymers ($\alpha = 25$ nM, same as that in the single-patch prism synthesis condition in Fig. 1, see Methods), assembling by their tips due to hydrophobic attraction between 2-NAT covered tips. Scale bar: 200 nm.

Comment 2: “For the colloidal bowties, in what conditions were they formed? What is the configuration of PAA after the merging of PS soft patches? Is it possible to assemble them into nanoprism-networks through patch merging of three-tip patched building blocks?”

Response: In Fig. 4, we show colloidal bowties are formed due to random collisions between polymer patches, interpenetrating to produce a uniform PS monomer concentration at the merged interface. In the original manuscript, the occasionally assembled bowties indeed have low yield.

In the revised manuscript, we added substantial new experimental results and presented a new assembly strategy based on surfactant-induced polyelectrolyte complexation (*Curr. Opin. Colloid Interface Sci.* **32**, 11 (2017); *Polymers* **11**, 51 (2019)). This is a method employed in making smooth and coherent polymer films (*Appl. Surf. Sci.* **458**, 903 (2018)) and in nucleotide purification (*Nat. Protoc.* **1**, 2320 (2006)) but has not been used in patchy particle assembly. It utilizes a positively charged surfactant molecule such as CTAB which can have their charged heads adsorbed and accumulated to the negatively charged PAA block of the polymer patch. The accumulated CTAB molecules lead to aggregation of the hydrophobic chains of CTAB, which bridges the patches of nearby NPs to drive directed assembly. This complexation effect is exclusive to polymer patches and does not cause prism aggregation via their sides and faces. This strategy has significantly increased the yield of selective, patch-to-patch only assembly to 80%. Furthermore, we apply this strategy to tri-patch prisms and indeed find networks as the reviewer expected.

In terms of the PAA conformation between the tips, we modeled the merging PAA blocks as a spherical packing of correlation blobs to fill up the space available between the tips. This picture is a manifestation of the idea that introducing charge moieties results in a reorganization of the charged PAA monomers into neutral correlation blobs that then interact via steric repulsion with each other. As discussed in our Supplementary Note 8, this assumption enables the usage of a classical semi-dilute scaling relationship, $N_x^{1/2} b_l \{b_l^3 c [v_o(1 - 2\chi)]^{-1}\}^{-1/8}$, between sterically interacting correlation blobs to model both the correlation size as well as the effective width of the merging regime between the two interacting tips.

To address the reviewer’s comments, we made the following revisions:

- (1) We added the description of the new assembly procedure and a new Fig. 5 in the main text and a new Supplementary Fig. 19 for more TEM images.

“To further increase the yield of selective, patch-to-patch assembly that enhances plasmonic coupling, we develop a new assembly method based on surfactant-induced complexation, which has been employed for polymer film formation⁵⁴ and nucleotide purification⁵⁵ but not for directing NP assembly. According to previous studies^{56,57}, cationic surfactants (e.g., CTAB) facilitate aggregation of anionic polyelectrolytes (e.g., PAA) by forming a surfactant-polyelectrolyte complex (Fig. 5a,b). The head group of CTAB can attach onto the polyelectrolytes by electrostatic attraction, followed by hydrophobic attraction among the tails driving the aggregation of polymers to which those tails are attached. Using this method for our single-patch prisms, we obtain 68% dimeric assemblies (376 structures counted, Fig. 5c–e, Supplementary Fig. 19) and 21% unassembled individual prisms. Importantly, among the dimeric assemblies, over 80% are bowties attached by the tip patches. Thus, this result shows greatly increased yield compared to the random patch merging shown in Fig. 4., implying patch-patch selective assembly.”

Figure 5. Patch-to-patch selective assembly of single-patch prisms. (a) Schematics of the CTAB-induced patch complexation mechanism. (b) Chemical structure of PS-*b*-PAA and CTAB involved in patch-patch selective assembly. (c) Low magnification TEM and SEM (inset) images of dimers. Yellow arrows highlight the merged patch interface. (d) High magnification TEM images of patch-patch assembled dimers. (e) Yields of assembly motifs in dimers. The notations in the *x*-axis mean as follows: n-n: between non-patched prism tips. p-s: patch to prism side. s-s: between prism sides. p-t: patch and non-patched tip. p-p: between patches. A total of 376 structures are counted. Inset: the yield of dimers compared to assemblies of trimers or larger (noted as '≥ 3') and non-assembled NPs (noted as '1'). Scale bars: 100 nm (c), 50 nm (d).

Supplementary Fig. 19. TEM images of CTAB-induced patch complexation of single-patched prisms into patch-patch selectively assembled dimers. Scale bars: 100 nm (a), 500 nm (c–e).

- (2) We also extended the strategy to the tri-patch prisms and indeed obtain networks as shown in the new Supplementary Fig. 21 and discussed the result in the main text.

“When we apply the same surfactant-induced complexation method to tri-patch prisms, we observe consistently tip-to-tip connected networks (Supplementary Fig. 21).”

Supplementary Fig. 21. Tri-patch prisms assembled into networks via the same CTAB-induced patch complexation during incubation in 50 nM CTAB solution. Scale bars: 100 nm.

- (3) Regarding the PAA conformation between the tips, we detail the model based on the polymer scaling theory and added the discussion above in Supplementary Note 8.

“Note that in terms of the PAA conformation between the tips, we model the merging PAA blocks as a spherical packing of correlation blobs to fill up the space available between the tips. This picture is a manifestation of the idea that introducing charge moieties results in a reorganization of the charged PAA monomers into neutral correlation blobs that then interact via steric repulsion with each other. As discussed above (Supplementary Note 8), this assumption enables the usage of a classical semi-dilute scaling relationship, $N_x^{1/2} b_l \{b_l^3 c [v_o(1 - 2\chi)]^{-1}\}^{-1/8}$, between sterically interacting correlation blobs to model both the correlation size as well as the effective width of the merging regime between the two interacting tips.”

Responses to Reviewer 3

Reviewer 3 commented that the authors “*presented an elegant procedure for patchy polymer-functionalized gold nanoparticles*” and “*this manuscript is of interest to the scientific community and could be published after the authors have addressed a few points.*” We thank the reviewer for the constructive comments to strengthen our manuscript, all of which we have addressed using new experiments and analysis (a new main text Fig. 5, five new supplementary figures: Supplementary Figs. 5, 12, 13, 19, 20, and updated Supplementary Tables 2, 3) as detailed below. In particular, we verified the general applicability of our strategy to achieve symmetry-breaking in two other nanoparticle shapes and two other polymers of different molecular weight. We also developed and demonstrated a new strategy to significantly increase the yield of patchy particle assembly.

Comment 1: “*The mechanism proposed by the authors supposed that the chemisorption of the NAT ligand occurs on all tips. Could this be quantified using EELS? Alternatively, can the gold nanoparticle be first functionalized with NAT and then PS-PAA after purification? The one-pot strategy, while practical, raise the question?*”

Response: We thank the reviewer for the suggestion! Although the use of EELS has been demonstrated to show a preferential coating of double-layered CTAB on the side over the tips of gold nanorods (*Nano Lett.* **19**, 6308 (2019)), it is still challenging to be used routinely to map the spatial distribution of a (sub)monolayer of short ligands (the 2-NAT). That quantitative mapping is beyond the scope of the work.

In the original manuscript, we proposed that the chemisorption of the 2-NAT ligand occurs on all tips based on our control experiments: (i) the adsorption behavior of 2-NAT on the gold NP surface does not change over temperature variations of 90–110 °C as suggested by our Raman characterization, “Raman spectra of the prisms show a constant peak intensity for the 2-NAT ligand³⁹ over the range of temperature (90–110 °C) (Methods, Supplementary Fig. 8a), suggesting negligible change in the 2-NAT ligand adsorption on NPs⁴⁰.” (ii) Over the entire 2-NAT concentration range we used in this work including that leads to single-patch prism synthesis (Fig. 1), at a grafting temperature of 110 °C, we consistently obtained tri-patch prisms, suggesting that this 2-NAT concentration range allows its adsorption onto all the three prism tips (Supplementary Fig. 2): “Notably, increasing to 110 °C induces symmetric grafting with a 99% yield of triple-patch NPs, consistent with our previous report³⁵.” “Therefore, the transition to triple-patch grafting at 110 °C is a result of weakened chain–chain attraction that produces equipartitioning of chains across all prism tips.” Thus, we conclude that lower temperature induces stronger polymer–polymer attraction which drives the formation of single- or double-patches. This observation is also *quantitatively* supported by polymer scaling theory and simulations. This conclusion is consistent with the fact that 2-NAT has smaller molecular weight (160 g/mol) and thus smaller intermolecular attraction in good solvent condition than the extended polymer chains (20,000 g/mol).

In the revised manuscript, we further clarified this point and added new experiments (Supplementary Figs. 5, 13) as detailed in our response to Comment 2 of Reviewer 1. In particular, we followed the reviewer’s suggestion to perform a multiple-step experiment that also confirms the all-tip coating of the NAT thiol (Supplementary Fig. 5).

- (1) We performed new experiments to show the prisms are coated with 2-NAT at all tips at 90 °C. First, we obtained the prisms that are processed at the same condition as that used for the single-patch prism presented in Fig. 1, but without the polymers. We find that these prisms, when dried, can assemble tip-to-tip connected through all three tips (Supplementary Fig. 5b), suggesting a balance of tip-to-tip hydrophobic attraction (due to 2-NAT coating) and electrostatic repulsion due to the residue charged ligands on the prism facets. Second, using these 2-NAT coated prisms, when we

then performed the subsequent experiment to incubate them with polymers, we still retrieved the single-patched prisms (Supplementary Fig. 5c), corroborating again that the asymmetric patch formation originated from the effect of polymer–polymer attractions.

Supplementary Fig. 5b,c. 2-NAT decorates all three tips. (b) SEM image of non-patch nanoprisms decorated with 2-NAT only, without polymers ($\alpha = 25$ nM, same as that in the single-patch prism synthesis condition in Fig. 1, see Methods), assembling by their tips due to hydrophobic attraction between 2-NAT covered tips. (c) Sequential addition of PS-*b*-PAA leading to asymmetric patch grafting despite 2-NAT covering all three tips. Scale bars: 200 nm (b), 100 nm (c).

(2) We added the discussions in the main text,

“In a sequential control experiment where the prisms are first incubated with 2-NAT, harvested, and then incubated with polymers, the prisms out of incubation with 2-NAT only can assemble into structures due to the hydrophobic attractions at all three tips (Supplementary Fig. 5b). Nevertheless, despite the 2-NAT decoration on all the three tips, the subsequent polymer grafting upon polymer addition generates single-patch prisms (Supplementary Fig. 5c), confirming that strong chain-chain interaction induces asymmetric grafting.”

Comment 2: “The authors claim that the strategy proposed is general. However, only one particle/polymer couple is studied. To strengthen their claim of a generalized strategy, the authors should investigate the formation of patches on other particles (nanorods?) and with at least one other copolymer.”

Response: We greatly appreciate the reviewer’s suggestion to demonstrate the generality of our approach. In the revised manuscript, we added substantial new experimental data on extending the strategy to other systems, including (i) *two more gold nanoparticle shapes* (octahedron, bipyramid); these two shapes have well-defined vertices in comparison with the rods suggested by the reviewer; and (ii) *two more polymer chains of different lengths* (PS₂₃₀-*b*-PAA₄₉, PS₃₉₄-*b*-PAA₅₈) in addition to the PS₁₅₄-*b*-PAA₅₁ we presented as the major dataset in the original manuscript. In all these systems, we are able to validate the mechanism of the supramolecular bandwagoning-induced asymmetric patch grafting and make asymmetrically patched nanoparticles. These new results are summarized as two new supplementary figures (Supplementary Figs. 12, 13), discussions in the main text, and details in Methods, supported by new simulations as detailed below.

(1) We extended the same strategy to differently shaped NPs: Among the 6 possible sites on the octahedron vertices, only 1–3 patches are grafted by the polymers when we do the grafting at 90 °C to enhance the polymer-polymer attraction (Supplementary Fig. 12a,b). For the bipyramid, among the 7 vertices (two at the long axis and five at the equator) only 1–3 patches are grafted by the polymers (Supplementary Fig. 12c,d). Because our modeling approach can already handle any

convex shape as inputs into both the theory and simulations, it was straightforward for us to perform the same set of calculations as described in the original manuscript for triangular prisms but now using an octahedron or bipyramid (Supplementary Fig. 12e). Our results reveal a similar preferential partitioning to the corners on both shapes. Sweeping across a subset of the full phase space explored for the triangular prism, we observe a similar transition from zero patch/sparse grafting to single/multiple distinct patches, to eventual patch merging. Due to the 3D nature of these shapes, patch merging along neighboring sides becomes more likely and emerges as a new phase regime. Full theoretical characterization of the phase boundary and modeling of patch merging, however, is beyond the scope of this work.

Supplementary Fig. 12. Extension to other NP shapes. (a,c) Asymmetric grafting on gold octahedra (a) and gold bipyramids (c) obtained at 90 °C. (b,d) Maps of 3D grafting probability p_{graft} and the representative simulated morphology of patchy octahedron (b) and bipyramids (d) obtained correspondingly. (e) Phase diagram of patch morphologies for octahedra and dipyramid. Ω here is a proxy for shapes to enable collapse of both sets of shape data into one phase diagram. Scale bars: 50 nm (a), 100 nm (c).

- (2) We performed new experiments on polymers of different lengths. Specifically, given a fixed 2-NAT concentration (α of 25 nM), for the new, intermediate-length polymer PS₂₃₀-*b*-PAA₄₉, single-patch prisms are obtained at a grafting temperature of 90 °C, while tri-patch prisms are dominant at a grafting temperature of 110 °C (Supplementary Fig. 13a,b), consistent with the behavior of the PS₁₅₄-*b*-PAA₅₁ polymer. By using even longer polymer chains PS₂₃₀-*b*-PAA₄₉, due to the stronger polymer–polymer attraction, single-patch prisms are obtained as the majority product even at 110 °C (Supplementary Fig. 13d). It is also consistent with the phase diagram on the x -axis toward the right direction as highlighted by the red arrow (Supplementary Fig. 13c).

Supplementary Fig. 13. Extension of the “bandwagon effect” to polymers of different chain lengths. (a,b) TEM images of single- and tri-patch nanoprisms obtained using PS₂₃₀-b-PAA₄₉ at $T = 90\text{ }^{\circ}\text{C}$ (a) and $T = 110\text{ }^{\circ}\text{C}$ (b) at a fixed 2-NAT concentration of $\alpha = 25\text{ nM}$. (c) Phase diagram of patch count n (shading colored to n as noted in the legend) across the parameter space: grafting temperature T , chain length N , and ligand concentration α . The phase diagram is partitioned to different colored regions following simulation results. The black lines are phase boundaries predicted by theory. Arrows show illustrations of estimated corresponding experimental variation in (a,b,d). (d) TEM images of single-patch nanoprisms obtained using PS₃₉₄-b-PAA₅₈ at $T = 110\text{ }^{\circ}\text{C}$ and $\alpha = 25\text{ nM}$. Scale bars: 100 nm.

- (3) We added the following discussion to the main text:
 “We verify the generalizability of the strong chain-chain interaction induced asymmetric grafting using differently shaped NPs and polymer chains of different lengths. On one hand, octahedron and bipyramid having one to three patches are formed, despite their number of vertices (i.e., 6 and 7, respectively, Supplementary Fig. 12). On the other hand, using PS₂₃₀-b-PAA₄₉, we see similar results as for PS₁₅₄-b-PAA₅₁. At a low grafting temperature of 90 °C, single-patch prisms are obtained whereas increasing temperature to 110 °C retrieves tri-patch prisms (Supplementary Fig. 13a,b). Interestingly, by using even longer polymer chains PS₂₃₀-b-PAA₄₉, we can get single-patch prisms even at 110 °C (Supplementary Fig. 13c,d). It is also consistent with the phase diagram (Fig. 2i) as one move towards the right along the x -axis.”

Comment 3: “Can the authors comment on the reversibility of the patches created? That is, if a particle with 1 patch prepared at low temperature, is purified and then incubated at a higher temperature will the morphology be preserved? Or if the solvent quality is changed?”

Response: To answer the reviewer’s questions, we performed new experiments to show single-patch prisms can convert to tri-patch prisms (that is, the patch created is reversible). In the revised manuscript,

(1) We added details of the experiment in the Methods section:

“For the reversibility test (Supplementary Fig. 5d,e), we kept the same reaction condition as for the single-patch prisms in Fig. 1b ($\alpha = 25$ nM and $T = 90$ °C). Once the reaction mixture in the vial is cooled down to room temperature, the same vial was brought into the 110 °C oil bath and left undisturbed for 2 h. The rest of the cooling, washing and sample preparation steps are the same as above for the one-step single-patch prism synthesis.”

(2) We added a new Supplementary Fig. 5d,e to summarize the results and added discussions in the main text,

“Furthermore, the single-patch prisms prepared at 90 °C, when incubated at 110 °C (Methods, Supplementary 5d,e), become tri-patched. This reversibility of patch formation is consistent with our hypothesis that the asymmetric and symmetric grafting is thermodynamically driven by chain–chain interaction.”

Supplementary Fig. 5d,e. (d) Single-patch prisms prepared at $T = 90$ °C and $\alpha = 25$ nM. (e) Tri-patch prisms obtained after heating up the single-patch prisms in (d) at 110 °C for another 2 h, showing the reversibility of asymmetric patch grafting. Scale bars: 100 nm (d,e), 50 nm (inset in e).

Comment 4: “Can the authors describe in more detail the formation of the bowtie assemblies and how the assembly process can be controlled. For example, does the concentration of NPs in suspension affect the fraction of NPs undergoing assembly?”

Response: In Fig. 4, we show colloidal bowties are formed due to random collisions between polymer patches, interpenetrating to produce a uniform PS monomer concentration within merged interface. Thus, if we increase the concentration of the nanoprism solutions, the fraction of NPs undergo assembly increases as well. However, the yield of colloidal bowties is still low, and most of the particles remain not assembled.

Thus, in the revised manuscript, going beyond the randomly assembled bowties, we presented a new strategy based on charged surfactant-induced polyelectrolyte complexation (*Curr. Opin. Colloid Interface Sci.* **32**, 11 (2017); *Polymers* **11**, 51 (2019)). This is a method employed in making smooth and coherent polymer films (*Appl. Surf. Sci.* **458**, 903 (2018)) and nucleotides purification (*Nat. Protoc.* **1**, 2320 (2006)) but has not been used in patchy particle assembly.

In the revised manuscript, we now provided a new main text Fig. 5, a whole new paragraph of discussion in the main text, new details in the “Method” section, and two new Supplementary Figures (Supplementary Figs. 19 and 20) to elaborate the details of this strategy. In summary, this strategy has significantly increased the yield of selective, patch-to-patch only assembly to 80%. For the mechanism and detailed explanation of the results as well as more control experiments (Supplementary Figs. 19–21), please refer to the answer to Comment 4 of Reviewer 1.

We added the description of the new assembly procedure and a new Fig. 5 in the main text, as well as a new Supplementary Fig. 19 for more TEM images.

“To further increase the yield of selective, patch-to-patch assembly that enhances plasmonic coupling, we develop a new assembly method based on surfactant-induced complexation, which has been employed for polymer film formation⁵⁴ and nucleotide purification⁵⁵ but not for directing NP assembly. According to previous studies^{56,57}, cationic surfactants (e.g., CTAB) facilitate aggregation of anionic polyelectrolytes (e.g., PAA) by forming a surfactant-polyelectrolyte complex (Fig. 5a,b). The head group of CTAB can attach onto the polyelectrolytes by electrostatic attraction, followed by hydrophobic attraction among the tails driving the aggregation of polymers to which those tails are attached. Using this method to our single-patch prisms, we obtain 68% dimeric assemblies (376 structures counted, Fig. 5c–e, Supplementary Fig. 19) and 21% unassembled individual prisms. Importantly, among the dimeric assemblies, over 80% are bowties attached by the tip patches. Thus, this result shows greatly increased yield compared to the random patch merging shown in Fig. 4., implying patch-patch selective assembly.”

Figure 5. Patch-to-patch selective assembly of single-patch prisms. (a) Schematics of the CTAB-induced patch complexation mechanism. (b) Chemical structure of PS-*b*-PAA and CTAB

involved in patch-patch selective assembly. (c) Low magnification TEM and SEM (inset) images of dimers. Yellow arrows highlight the merged patch interface. (d) High magnification TEM images of patch-patch assembled dimers. (e) Yields of assembly motifs in dimers. The notations in the x -axis mean as follows: n-n: between non-patched prism tips. p-s: patch to prism side. s-s: between prism sides. p-t: patch and non-patched tip. p-p: between patches. A total of 376 structures are counted. Inset: the yield of dimers compared to assemblies of trimers or larger (noted as ' ≥ 3 ') and non-assembled NPs (noted as '1'). Scale bars: 100 nm (c), 50 nm (d).

Comment 5: “Can the authors provide the number of particles analyzed in tables 2-3 in supporting information.”

Response: In the original manuscript, the numbers of particles analyzed in Supplementary Tables 2 and 3 were listed in the table captions. In the revised manuscript, we now provide the numbers of particles analyzed in each row in the tables as below.

Supplementary Table 2. Fractions of patchy nanoprisms with varied patch count n obtained at different temperature T at fixed α of 50 nM as shown in Fig. 2e. The “Other” refers to prisms with at least one prism side fully coated.

T (°C)	Zero-patch	Single-patch	Double-patch	Triple-patch	Other	Patch count per prism n	Standard deviation of n	Number of particles analyzed
90	0	0.136	0.673	0.164	0.028	1.97	0.55	214
100	0	0.105	0.250	0.640	0.006	2.52	0.68	172
110	0	0	0.010	0.985	0.005	2.98	0.10	200

Supplementary Table 3. Fractions of patchy nanoprisms with varied patch count n obtained at different α and fixed temperature T of 90 °C as shown in Fig. 2g. The “Other” refers to prisms with at least one prism side fully coated.

α (nM)	Zero-patch	Single-patch	Double-patch	Triple-patch	Other	Patch count per prism n	Standard deviation of n	Number of particles analyzed
25	0.091	0.818	0.085	0	0.006	0.99	0.42	330
37.5	0.009	0.462	0.417	0.085	0.027	1.55	0.65	223
75	0	0.051	0.328	0.610	0.011	2.54	0.59	177

REVIEWERS' COMMENTS

Reviewer #1 (Remarks to the Author):

The authors have substantially improved the manuscript with experiments and explanations and have removed any previous concerns from a scientific point of view. I recommend publication in Nature Communications.

As mentioned in the previous review round, please reconsider the use bandwagoning or at least rewrite passages like: "...bandwagon effect, whereby individuals adopt particular behaviors, mannerisms or views simply because others have adopted them." The manuscript shows plain and simple polymer/particle aggregation and no conscious decision making.

Reviewer #2 (Remarks to the Author):

The authors have done very nice work and went beyond to respond my comments. I would recommend to publish the work as it is.

Reviewer #3 (Remarks to the Author):

In the revised version of Symmetry-breaking in patch formation on gold nanoparticles via supramolecular "bandwagoning", the authors have added clarified their discussion and added new results to fully support their claims.

In this new version, the authors have fully addressed my previous concerns. The paper, in its revised version, reports highly compelling results that are strongly backing the analysis and discussion.

I believe the paper should be published in its current form.